

# Redescription of *Calyptosuchus* (*Stagonolepis*) *wellesi* (Archosauria: Pseudosuchia: Aetosauria) from the Late Triassic of the Southwestern United States with a discussion of genera in vertebrate paleontology

William G. Parker

Division of Science and Resource Management, Petrified Forest National Park, Petrified Forest, AZ, USA

Corresponding author
William G. Parker,
William_Parker@nps.gov

## ABSTRACT

*Calyptosuchus wellesi* is a medium-sized desmatosuchian aetosaur common in Adamanian (early to middle Norian) age rocks from the Chinle Formation and Dockum Group of the Western United States. Known chiefly from osteoderms, this taxon has never been fully described and non-osteoderm material assigned to *Calyptosuchus* has been done so based on questionable criteria. Mapping of aetosaurian elements from the *Placerias* Quarry allows for the recognition of associated material providing support for referrals of non-osteoderm material. Furthermore, another previously undescribed specimen from the Chinle Formation of Arizona provides more details about this taxon. Presently *Calyptosuchus* lacks discrete autapomorphies, but can be distinguished from other aetosaurs based on a unique combination of characters supported by a phylogenetic analysis. *Calyptosuchus* is one of the most common aetosaurians in the Western United States and an index taxon of the early Adamanian biozone. The name *Calyptosuchus* is retained and encouraged as the applicable genus name for the species *wellesi* rather than the often used *Stagonolepis* because assignments of taxa to multi-species genus names are problematic and in this case provides a proposed taxonomic relationship that cannot be unambiguously supported, even by phylogenetic analyses. Because of the inherent limitations of the fossil record, referral of specimens and species to species and genera respectively is an epistemological problem in vertebrate paleontology.

## INTRODUCTION

Aetosaurs are quadrupedal, armored, possibly herbivorous archosaurs known exclusively from Late Triassic deposits throughout Pangea (*Desojo et al., 2013*). The most commonly recovered fossils of aetosaurs are their characteristic osteoderms, which can be diagnostic

to various taxonomic levels including species and are the basis for phylogenetic studies of the group (*Desojo et al., 2013*; *Parker, 2016a*). Presently there are 17 valid species of aetosaur known from North America.

In 1931 Ermine Cowles Case of the University of Michigan Museum of Paleontology (UMMP) discovered a well-preserved articulated partial carapace with an associated vertebral column and pelvis of an aetosaurian in Upper Triassic strata of the Texas Panhandle. Although described in detail, the taxonomic affinities of the specimen at the time were considered enigmatic and the material was assigned only to Phytosauria (*Case, 1932*).

That same year Charles Lewis Camp of the University of California Museum of Paleontology (UCMP) began excavating a vast deposit of bones in the Upper Triassic Chinle Formation of Arizona at a site he christened the *Placerias* Quarry because of the large number of bones of the dicynodont *Placerias gigas* (=*Placerias hesternus*) recovered there (*Camp & Welles, 1956*). In addition, Camp recovered a large number of aetosaurian "skin plates" (his term for osteoderms) as well as endoskeletal (non-osteoderm) portions of the skeletons of dozens of individuals (*Long & Murry, 1995*). Comparison of this material to that of *Stagonolepis robertsoni* from the Elgin Sandstone (now the Lossiemouth Sandstone Formation) of Scotland led Camp to believe that much of his Arizona material represented a very similar animal, possibly of the same genus (C. L. Camp, 1935, unpublished data). Unfortunately Camp never published descriptions or taxonomic notes regarding these specimens, only referring them in passing to "*Typothorax*" (as in *Longosuchus meadei*) and "*Episcoposaurus*" (as in *Desmatosuchus spurensis*) (*Camp & Welles, 1956*: 259).

Both the Texas and Arizona material remained undescribed until it was restudied as part of a field investigation of the Triassic of Arizona by crews from the UCMP in the 1980s (*Long & Ballew, 1985*). During this time it was named *Calyptosuchus wellesi* and Case's specimen was designated as the holotype of this new taxon (*Long & Ballew, 1985*). The generic name was only used for a very short time before it was noted again that the material appeared to be very similar to that of *S. robertsoni*, and was reassigned to the genus *Stagonolepis*, as *Stagonolepis wellesi* (*Murry & Long, 1989*). *S. wellesi* was differentiated from *S. robertsoni* by the presence of short horns on the cervical lateral osteoderms (*Long & Ballew, 1985*; *Long & Murry, 1995*); however, these were later demonstrated to belong to a previously unrecognized paratypothoracin aetosaur that was present in the *Placerias* and Downs quarries at St. Johns Arizona, probably *Tecovasuchus* (*Parker, 2005*; *Heckert et al., 2007*). Thus, specific characters that diagnose *S. wellesi* sensu *Long & Murry (1995)*, exclusive of other aetosaurians, are lacking. Initial comparisons of the dorsal osteoderms with those of *S. robertsoni* for this study revealed strong differences (see Discussion below) and the use of *C. wellesi* for the North American material is recommended (*Parker, 2008a*; *Parker & Martz, 2011*; *Desojo et al., 2013*).

Scoring *C. wellesi* into a phylogenetic analysis is challenging because the holotype consists of the articulated carapace from just anterior to the pelvic region back through the middle of the tail, and it lacks both limb and cranial material. Furthermore, the specimen was set in plaster and mounted upright behind heavy glass in the UMMP.

The associated vertebral column and pelvis were separated from the osteoderms during the initial study by Case and are presently in poor condition (W. Parker, 2000, personal observation).

Besides *Case's (1932)* description of UMMP 13950 and his descriptions of a referred isolated pelvis and associated vertebrae (UMMP 7470; *Case, 1922*, *1929*), *C. wellesi* has never been adequately described. The initial study in which the taxon was named only provides a general list of characters of the osteoderms (*Long & Ballew, 1985*). Superficial descriptions of various referred endoskeletal elements were provided by *Long & Murry (1995)*, who did not redescribe the type or referred osteoderms in more detail.

In this paper field collection numbers are used to try to recover associations between the diagnostic osteoderms of *C. wellesi* and other elements of the skeleton which are redescribed in the modern context of our understanding of aetosaurian anatomy. The referral of this material to the genus *Calyptosuchus* rather than *Stagonolepis* is controversial, so the rationale behind this assignment is discussed as is the problem of the genus-group taxonomic rank in Triassic vertebrate paleontology.

## Geological setting of the *Placerias* Quarry

The *Placerias* Quarry is situated in a small area of badlands in Apache County, Arizona southwest of the city of St. Johns (Fig. 1A). These outcrops represent the Upper Triassic Chinle Formation (*Akers, 1964*) and the quarry itself is developed in an olive gray claystone lens with abundant carbonate nodules (*Fiorillo, Padian & Musikasinthorn, 2000*). The quarry has been interpreted as a stagnant waterhole or bog (*Camp & Welles, 1956*), but a more recent study found the local sedimentology to be consistent with pedogenically modified fluvial sediments in an area with seasonally high water tables and periods of aridity (*Fiorillo, Padian & Musikasinthorn, 2000*).

The stratigraphic position of the quarry is controversial because of poor exposure of the outcrops (*Fiorillo, Padian & Musikasinthorn, 2000*), but all authors agree that it occurs in the lower portion of the Chinle Formation (*Camp & Welles, 1956*; *Jacobs & Murry, 1980*; *Long & Murry, 1995*; *Lucas, Heckert & Hunt, 1997*; *Fiorillo, Padian & Musikasinthorn, 2000*; *Parker & Martz, 2011*). Lithostratigraphic correlation from Petrified Forest National Park, approximately 62 km northwest of the quarry, demonstrates that the quarry is located either in the uppermost portion of the Blue Mesa Member or in the lowermost portion of the Sonsela Member (*Parker & Martz, 2011*; *Irmis et al., 2011*; W. G. Parker & J. W. Martz, 2010, unpublished data). Redbeds above the quarry level assigned to the Bluewater Creek Member of the Chinle Formation (*Lucas, Heckert & Hunt, 1997*) are actually deposits of the Miocene–Pliocene Bidahochi Formation separated from the Chinle Formation by angular unconformities (*Akers, 1964*; W. G. Parker & J. W. Martz, 2008, unpublished data). The maximum depositional age of the quarry is established by high-precision U–Pb geochronology to be 219.39 ± 0.16 Ma (*Ramezani, Fastovsky & Bowring, 2014*). This would make it equivalent in age to the upper part of the Lot's Wife beds of the lower Sonsela Member at PEFO (Fig. 1B; *Martz & Parker, 2010*; *Ramezani et al., 2011*; *Atchley et al., 2013*). The quarry is in the Adamanian Teilzone (*Martz & Parker, 2017*).

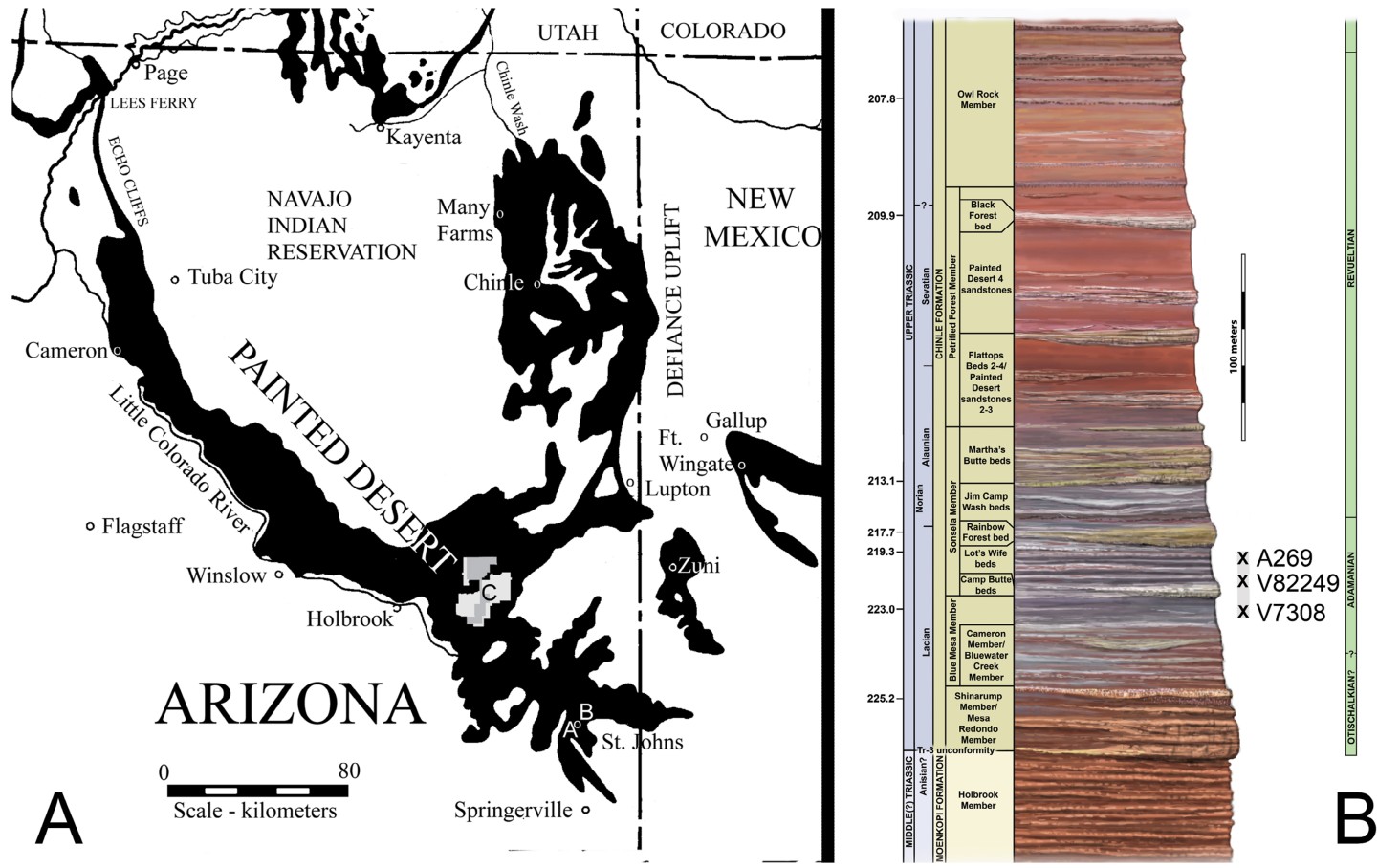

**Figure 1 Locality map and stratigraphic section.** (A) Map of Chinle Formation localities in northeastern Arizona; A. *Placerias* Quarry; B. Blue Hills; C. Petrified Forest National Park. (B) Stratigraphic section of the Chinle Formation near Petrified Forest National Park, showing the position of the localities from (A) and the stratigraphic range of *Calyptosuchus wellesi*. Stratigraphy from *Martz et al. (2012)*. Radioisotopic dates from *Ramezani et al. (2011)*. Relative position of locality V7308 from J. W. Martz & W.G. Parker (2008, unpublished data). Position of V82249 from *Parker & Martz (2011)*. Position of A269 based on geochronological correlation using data from *Ramezani, Fastovsky & Bowring (2014)*.

## MATERIALS AND METHODS

### *Calyptosuchus* material from the *Placerias* Quarry

The largest collection of material referred to *C. wellesi* is from the *Placerias* Quarry (UCMP A269/MNA 207-1) and potentially contains bones from most portions of the skeleton including a few isolated skull bones and basicrania (see below). *Long & Murry (1995)* referred much of this material to *Calyptosuchus*; however, many of these elements have received unique catalogue numbers and any original association has been lost. Furthermore, *Camp & Welles (1956)* stated that little of the material in the quarry was associated. Thus, it is not clear on what basis the endoskeletal material was assigned to *Calyptosuchus* by *Long & Murry (1995)*. However, several disarticulated archosaur specimens from the quarry articulate together, demonstrating that they belong to the same individual. The best example from the quarry are five elements (UCMP 25962, right ilium, UCMP 25974, left ilium, UCMP 25999, pubis, UCMP 25993, ischium,

UCMP 78719, sacral vertebrae), which can be combined to reconstitute a nearly complete pelvis of *Poposaurus gracilis* (*Long & Murry, 1995*: figs. 151, 153). The quarry also contains associated pelvic and limb material from a single individual of *C. wellesi* (*Long & Murry, 1995*: fig. 79), which is discussed in more detail below.

Fortunately, the collectors at the *Placerias* Quarry excavated utilizing a grid system (*Camp & Welles, 1956*) and physically marked the grid of collection in permanent ink on many of the bones. These numbers can be matched to the published quarry map (*Camp & Welles, 1956*: fig. 2), and although the exact placements within the grid for each bone have not been preserved, the numerous smaller grids measure about 2.25 m$^2$ and the largest about 9 m$^2$ (*Camp & Welles, 1956*), allowing for some degree of association to be estimated. With the exception of a few endoskeletal elements discussed in the text, only the osteoderm material can be assigned with any certainty to the genera *Calyptosuchus* and *Desmatosuchus*. For this study a spreadsheet was created listing all of the material (over 900 specimens) assigned to these taxa by *Long & Murry (1995)* along with the associated field/grid number (Supplemental Information). The element types were then plotted onto the quarry map with the exception of the majority of the numerous caudal centra, which are indeterminate to genus or species (Fig. 2). No other aetosaurians were recognized in the plotted osteoderm sample even though rare paratypothoracin lateral osteoderms are recognized from collections from the area made at later dates (*Parker, 2005*). Thus, all of the material is considered referable to *Calyptosuchus* or *Desmatosuchus* with the caveat that the slight possibility does exist that some of the endoskeletal elements could represent the extremely rare paratypothoracin that is known from armor from the nearby Downs Quarry.

Plotting the sorted data shows large accumulations of *C. wellesi* osteoderms in grids C71S and C72S, as well as in C64M and C65M (Fig. 2). *D. spurensis* osteoderms are accumulated particularly in C75W, C64, and C62M (Fig. 2). Thus, there is some distinction between large accumulations of osteoderms of these taxa and it is possible that these associations could represent single individuals. This information is used to make suggestive referrals of material to *C. wellesi* and is discussed in more detail in the following description. Unfortunately there is no way to calculate a genuine minimum number of individuals for each taxon; however, there are 14 aetosaurian basicrania in the overall sample (including three that lack field numbers). Numerous endoskeletal elements in CD1, CD2, CE1, CE2, CF1, CF2 are associated with very few osteoderms presenting a potentially interesting taphonomic question of why they are lacking; however, *Camp & Welles (1956*: 259*)* note that in this portion of the excavation "most of the numerous isolated dermal scutes of *Typothorax*, as well as broken ribs and other fragmentary material, were not collected." Thus the majority of osteoderms in the *Placerias* Quarry sample were collected in 1931 from the west side of the quarry and in 1932, during excavation of the east side, the osteoderms were ignored. This is reflected in the plotted data (Fig. 2). Note that by listing "*Typothorax*," *Camp & Welles (1956)* were actually referring to *Calyptosuchus*, although they are may also be using this name to encompass all of the aetosaurian paramedian osteoderms.

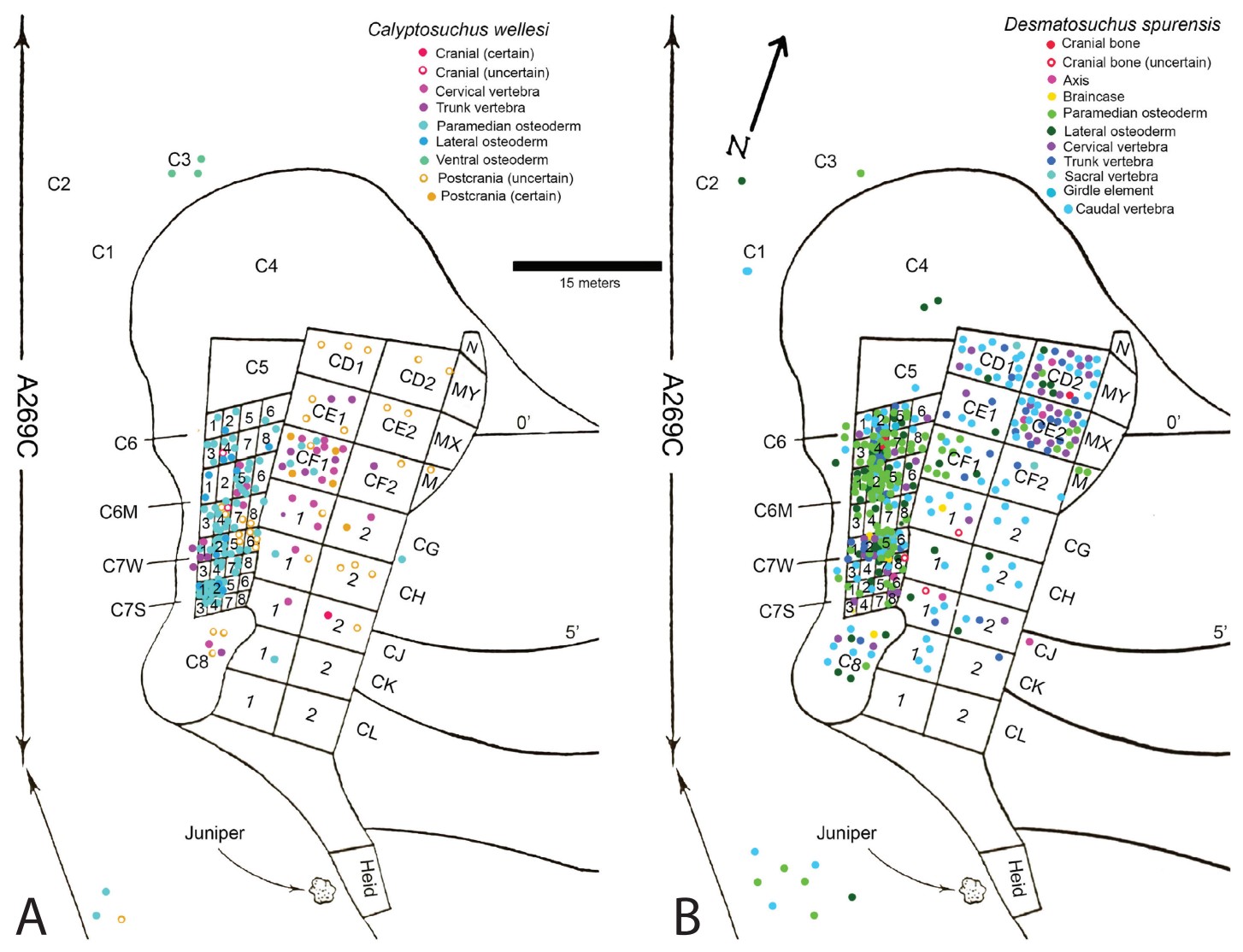

**Figure 2 Aetosaur elements plotted on quarry map.** Recovered elements of (A) *Calyptosuchus wellesi* and (B) *Desmatosuchus spurensis* plotted on the map of the *Placerias* Quarry. Map redrawn and modified from *Camp & Welles (1956)*.

# SYSTEMATIC PALEONTOLOGY

Archosauria *Cope, 1869* sensu *Gauthier & Padian, 1985*

Pseudosuchia *von Zittel, 1887–90* sensu *Gauthier & Padian, 1985*

Aetosauria *Marsh, 1884* sensu *Parker, 2007*

Desmatosuchia *Case, 1920* sensu *Parker, 2016a*

Desmatosuchinae *Case, 1920* sensu *Heckert & Lucas, 2000*

*Calyptosuchus Long & Ballew, 1985*

*Calyptosuchus wellesi Long & Ballew, 1985*

(Figs. 3–19)

*1922*   Phytosaur: Case, p. 73, fig. 28b.

*1929*   Phytosaur: Case, p. 49, fig. 21.

*1932*   *Phytosaurus*?: Case, p. 57, figs. 1–6, pl. 1–3, pl. 4, fig. 1.

*1953*   *Typothorax*: Gregory, p. 13.

*1953*   *Desmatosuchus haplocerus*: Gregory, p. 15.

*1961*   Unnamed aetosaur: Walker, p. 157

*1961*   *Desmatosuchus haplocerus*: Walker, p. 181.

*1961*   *Typothorax*: Walker, p. 184.

*1962*   *Phytosaurus*: Gregory, p. 682.

*1985*   *Calyptosuchus wellesi*: Long & Ballew, p. 47, figs. 13b, 14b, 15–16, pl. 4–5. [non fig. 13a, 14a (= *Scutarx deltatylus*)].

*1986*   *Calyptosuchus*: Long & Padian, p. 165.

*1986*   *Calyptosuchus*: Parrish & Carpenter, p. 158.

*1986*   *Calyptosuchus wellesi*: Murry, p. 123.

*1988*   *Calyptosuchus wellesi*: Long & Houk, p. 50.

*1989*   *Stagonolepis wellesi*: Murry & Long, p. 32.

*1995*   *Stagonolepis wellesi*: Long & Murry, p. 1, figs. 68–70, 71a, c, d, 72a, c–d, f–g, 73–77, 79–81, 83–84. [non figs. 71b, 72b, e (=*Scutarx deltatylus*), 71e–f (=Paratypothoracini), 78, 82 (=Stagonolepididae)].

*1996a*   *Stagonolepis wellesi*: Lucas & Heckert, p. 70.

*1996b*   *Stagonolepis wellesi*: Lucas & Heckert, p. 60, fig. 4 (in part). [non fig. 4 (in part) (=*Scutarx deltatylus*)].

*1997*   *Stagonolepis*: Heckert & Lucas, p. 14.

*1997*   *Stagonolepis wellesi*: Lucas, Heckert & Hunt, p. 40.

*1998*   *Stagonolepis wellesi*: Lucas, p. 366, fig. 11b (in part). [non fig. 11b (in part) (=*Scutarx deltatylus*).

*2000*   *Stagonolepis wellesi*, Heckert and Lucas, p. 1543, figs. 4a–b

*2002*   *Stagonolepis wellesi*, Heckert and Lucas, p. 12.

*2005*   *Stagonolepis wellesi*: Heckert, Lucas & Hunt, p. 23.

*2005*   *Stagonolepis wellesi*: Parker, p. 38.

*2005*   *Stagonolepis wellesi*: Parker & Irmis, p. 50. [non fig. 4a (=*Scutarx deltatylus*)].

*2005*   *Stagonolepis wellesi*: Irmis, p. 77, fig. 6e.

*2006*   *Stagonolepis wellesi*: Parker, p. 47.

*2007*   *Stagonolepis wellesi*: Parker, p. 54.

*2008*   *Desmatosuchus haplocerus*: Lucas & Connealy, p. 26.

*2010*   *Stagonolepis*: Lucas, p. 464.

*2011*   *Calyptosuchus wellesi*, Parker & Martz, p. 240, fig. 3.

*2013*   *Calyptosuchus wellesi*: Desojo et al., p. 206.

2013    *Calyptosuchus wellesi*: Martz et al., p. 346. [non figs. 7a–d (=*Scutarx deltatylus*)].

2016a    *Calyptosuchus wellesi*: Parker, p. 2, fig. 24a.

2016b    *Calyptosuchus wellesi*: Parker, p. 13.

**Holotype:** UMMP 13950, partial articulated skeleton consisting of the osteoderms of the posterior dorsal series through the mid-caudal region, the associated partial vertebral column and the sacrum (*Case, 1932*).

**Referred Specimens:** UMMP 7470, mostly complete pelvis with associated posterior trunk vertebrae and paramedian osteoderms from the Tecovas Formation near Holmes Creek in Crosby County, Texas (*Case, 1922*); UCMP 27225, dentary fragment, dentigerous bone fragment, cervical centra, paramedian, lateral, and ventral osteoderms from the Blue Hills, St. Johns, Arizona (UCMP loc. V7308; Figs. 1A and 1B); UCMP 126844, 10 paramedian osteoderm fragments from Petrified Forest National Park, Arizona (UCMP loc. V82249, PFV 162; Figs. 1A and 1B). Much material from the *Placerias* Quarry (UCMP loc. A269; Figs. 1A and 1B) near St. Johns, Arizona is referable to *C. wellesi* as is other material from Petrified Forest National Park (*Long & Murry, 1995*; *Parker & Martz, 2011*; see description below).

**Stratigraphic Horizon and Age:** Upper part of the Blue Mesa Member and lower part of the Sonsela Member (sensu *Martz & Parker, 2010*), Chinle Formation, Arizona (Fig. 1B); Tecovas Formation, Dockum Group, Texas. Adamanian Estimated Holochronozone and Estimated Holochron (224–215 Ma; *Martz & Parker, 2017*), early Norian (*Furin et al., 2006*).

**Revised Diagnosis:** Medium-sized (less than 4 m in total length) aetosaur that presently lacks discrete autapomorphies, but differs from other aetosaurs based on a unique combination of characters: large knob-like dorsal eminences that contact the posterior margin of the dorsal and caudal paramedian osteoderms; moderate width/length ratios of the dorsal trunk paramedian osteoderms; strongly radial pattern of ridges and furrows on paramedian osteoderms; anterolateral and anteromedial projections of the anterior bar of the paramedian osteoderms as in non-desmatosuchins; triangular projection of the anterior bar anterior to the dorsal eminence on the dorsal trunk paramedian osteoderms; dorsal paramedian osteoderms with a "scalloped" anterior margin of the anterior bar between the medial edge and the anterior triangular projection; dorsal trunk paramedian osteoderms with a weak ventral strut; cervical vertebrae are keeled ventrally; trunk vertebrae lack hyposphene–hypantrum articulations; base of the postzygapophyses of the trunk vertebrae bearing a posterior projection that rests upon the ventral bar of the prezygapophyses; neural spines taller than the centra in the mid-trunk vertebrae; posterior end of the iliac blade squared off; dentary with nine tooth positions. Differs from *S. deltatylus* in that the cervical and dorsal trunk paramedian osteoderms lacks a pronounced triangular protuberance in the posterolateral corner. Differs from *Aetosauroides scagliai* in possessing a dentary that bears a sharp inflexion on the ventral margin. Differs from *Adamanasuchus eisenhardtae* in that the trunk paramedians lack the "cut-off" posterolateral corners found in *Adamanasuchus*. Differs from *S. robertsoni* in

possessing transversely oval, instead of circular, articular faces of the cervical vertebrae; ventrally opening acetabulae; a squared off posterior end of the iliac blade; and an elongate anterolateral projection of the anterior bar on the trunk paramedian osteoderms. *S. robertsoni* also appears to lack the weak ventral strut on the paramedian osteoderms.

## DESCRIPTION

### Cranial bones

From a partial skeleton with osteoderms the only skull bone unambiguously referable to *C. wellesi* is a partial right dentary from UCMP 27225, which was neither mentioned nor described by *Long & Ballew (1985)* or *Long & Murry (1995)*. This partial dentary is missing all of the anterior portion as well as the posterior articulations with the angular and surangular (Fig. 3A). The element is slightly crushed and still covered in part by a hematite crust, but many details can be discerned. Overall the element is dorsoventrally shallow and possesses the sharp inflexion on the ventral margin of the dentary described by *Desojo & Ezcurra (2011)* as present in *Desmatosuchus smalli*, *S. robertsoni*, and *Neoaetosauroides engaeus*, and as lacking in *A. scagliai*. The medial surface is inscribed by an elongate, tapering Meckelian groove, which extends anteriorly to the level of the third alveolus (Fig. 3B). The anteroventral corner of the medial surface bears a rugose patch that represents the beginning of the dentary symphysis. The occlusal surface is slightly concave, edentulous anteriorly and preserving nine oval alveoli posteriorly. The alveoli are closely spaced and slightly imbricated (Fig. 3C). No complete teeth are preserved although root fragments are present in some of the alveoli. A second dentigerous fragment in UCMP 27225 bears five alveoli and represents a portion of the maxilla.

There are numerous aetosaur frontals and parietals in the UCMP collection from the *Placerias* Quarry, but none can be referred with certainty to *Calyptosuchus*. There are also approximately nine basicrania in the same collections. Two (UCMP 27414, UCMP 27419) possess anteroposteriorly elongate basisphenoids with divergent basipterygoid processes. These differ significantly from those of *Desmatosuchus* (TTU P-9023; UMMP 7476) and may belong to *Calyptosuchus*; however, this cannot be presently ascertained.

There are also two maxillary fragments that also differ in morphology from known specimens of *Desmatosuchus* (e.g., TTU P-9024; UMMP 7476) in possessing a distinct antorbital fossa delineated ventrally and anteriorly by a sharp rim (Fig. 3). The first (UCMP 195193) is a fragment of a right maxilla which preserves the main body ventral to the anterior portion of the antorbital fossa including the base of the ascending process of the maxilla (Figs. 4A–4C). The lateral face is divided into two sections by a sharp horizontal ridge that forms the ventral border of the antorbital fossa. Anteriorly this ridge forms a broad dorsally sweeping curve that extends up onto the ascending process of the maxilla. A similar ridge is present in *Stagonolepis olenkae* (*Sulej, 2010*), *A. scagliai* (PVL 2073), *S. robertsoni* (*Walker, 1961*), and *Revueltosaurus callenderi* (PEFO 34561), but is absent or extremely weak in *Desmatosuchus* (e.g., TTU P-9024) and *L. meadei* (TMM 31100-98). In *S. olenkae* the ventral portion is not as deep and as a result the ridge does not split the main body of the maxilla in two equal portions. This maxillary fragment

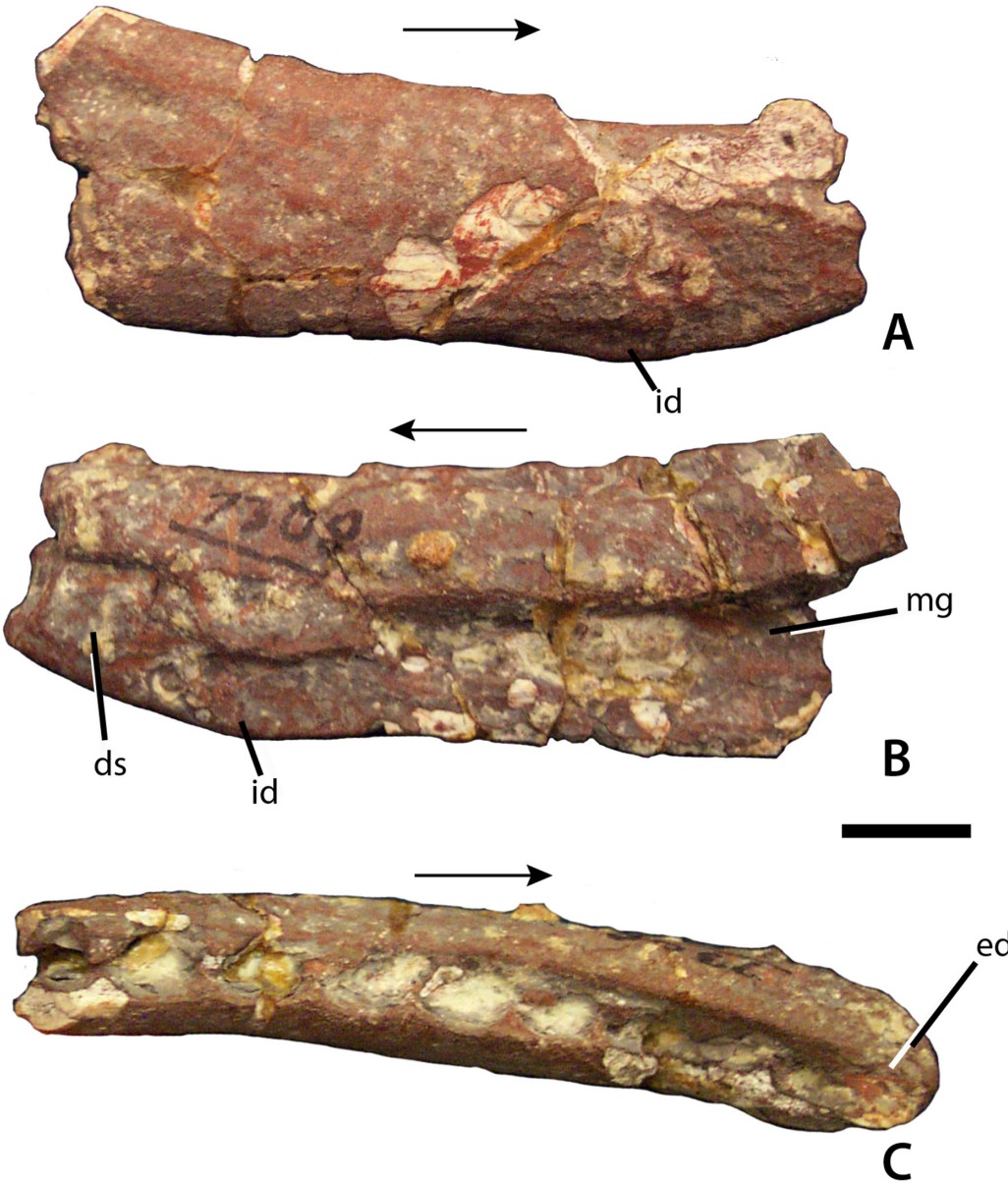

**Figure 3 Dentary of *Calyptosuchus wellesi*.** Partial right dentary of *Calyptosuchus wellesi* (UCMP 27225) in lateral (A), medial (B), and occlusal (C) views. Scale bar = 1 cm. Arrows indicate anterior direction. ds, dentary symphysis; ed, edentulous area; id, dentary infexion; mg, Meckelian groove.

is missing the anterior and posterior portions as well as the majority of the ascending process and as preserved has a length of 45.7 mm and a height of 36.8 mm. The height from the ventral margin to the antorbital fenestra is 18.2 mm. The margin of the antorbital fenestra is thin. The fenestra was longer than high, and ovate in outline. The contact with the nasal is preserved as a shallow, concave groove with a sharp, medial ridge (s.na, Figs. 4A and 4B). In lateral view this groove slopes anteroventrally.

In ventral view the anterior portion of the maxillary fragment is mediolaterally crushed. Four complete and part of a fifth alveoli are preserved. The third alveolus

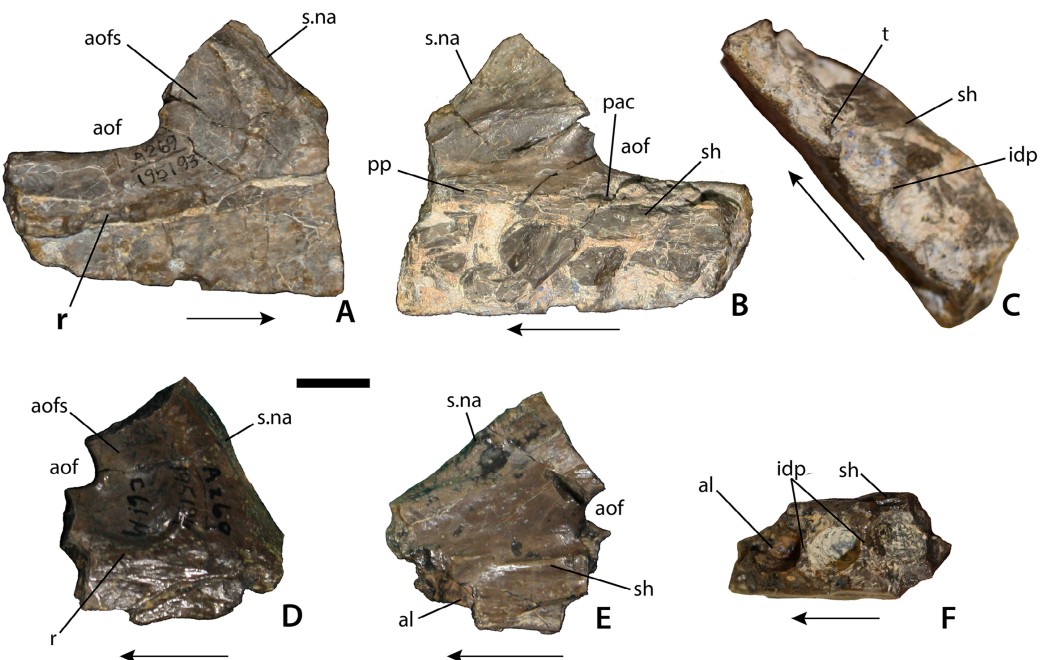

**Figure 4 Maxilla of *Calyptosuchus wellesi*.** Maxillary fragments possibly referable to *Calyptosuchus wellesi*. (A–C) Right maxilla (UCMP 195193) in lateral (A), medial (B), and occlusal (C) views. (D–F) Right maxilla (UCMP 195194) in lateral (D), medial (C), and occlusal (F) views. Scale bar equals 1 cm. Arrows indicate anterior direction. al, alveolus; aof, antorbital fenestra; aofs, antorbital fossa; idp, interdental plate; pac, pneumatic accessory cavity; pp, palatal process of the maxilla; r, ridge; sh, maxillary shelf; s.na, suture with the nasal element; t, tooth.

(from the front) preserves an unerupted tooth, but no further details can be made out. Interdental plates are present, but unfused (Fig. 4C). Medially there is a transverse ridge above the tooth row for articulation with the palate and forms a broad shelf bordering the antorbital fenestra (sh, Fig. 4B). There is a marked foramen (corresponding to the pneumatic accessory cavity of *Small (2002)*) at the anteroventral corner of the antorbital fenestra, which is visible medially and dorsally. The anterior portion of the maxillary body is concave and a small ridge marks about where the upper border of the antorbital fenestra would be located. Dorsal to this is another smooth concave area.

The second specimen (UCMP 195194) is also from the right side and therefore from a different individual (Figs. 4D and 4E). The anterolateral surface below the antorbital fossa is slightly rugose. The "pneumatic accessory cavity" (*Small, 2002*) is visible in medial view and has possibly been enlarged by preparation. Anteriorly the nasal articulation is preserved and similar to the first specimen. Anterior to this is a thin rim of bone that represents the posteroventral margin of the external naris. Thus the maxilla enters the naris, differing from the condition in *A. scagliai* (PVL 2073), where a thin contact of the premaxilla and the nasal exclude the maxilla from the margin of the external naris (*Casamiquela, 1961*; *Desojo & Ezcurra, 2011*). On the medial surface, a sharp raised ridge is preserved anteriorly that represents the palatal process of the maxilla. Only three alveoli are preserved in this fragment.

Despite the strong possibility of these cranial elements belonging to *C. wellesi*, they should not be used to score phylogenetic characters until they can be assigned with absolute certainty.

## Postcrania

### *Atlas/axis*

There are many axes in the collection from the *Placerias* Quarry. *Case (1922)* describes the ventral surface of the axis in *D. spurensis* as flat, and most of the specimens in the collection possess flat ventral surfaces. However, UCMP 139803 (from CF1) has a distinct ventral keel (Fig. 5A) and therefore mostly likely is referable to *C. wellesi* which has keeled cervical vertebrae (e.g., UCMP 27225; *Murry & Long, 1989*) rather than *D. spurensis* which has cervicals with a smooth ventral surface (e.g., UMMP 7476; MNA V9300). The upper portion of the neural arch, including the zygapophyses, is broken (Figs. 5B–5D). The atlantal neural arches are also broken. The centrum of the axis has distinct concave sides that are overhung by a thickened ridge, which bears the diapophyses (Fig. 5D). The parapophyses are situated anteroventrolaterally on the centrum and are connected ventrally by a thickened crescentic ridge that forms the anterior portion of the atlas intercentrum (Fig. 5A). The suture between the atlas intercentrum and the axis centrum is visible in ventral view.

The parapophyseal facets are round and directed ventrolaterally and slightly posterior. The odontoid process is attached (Figs. 5A, 5B, 5D and 5E); its sutures with the centrum are still visible so the fusion is not complete. The dorsal surface of the odontoid process forms a slightly concave trough that opens posteriorly into the neural canal (Fig. 5E). The canal is large, about one-half the diameter of the posterior articular face of the centrum. In posterior view, the articular face of the centrum has a flat (horizontal) dorsal margin. The face is concave with well-developed rims. The length of the atlas/axis including the odontoid process is 48.7 mm. The axis centrum has a width of 30.6 mm and a height of 25.4 mm.

### *Postaxial cervical vertebrae*

Numerous vertebrae were recovered in grid square CF1, where the atlas/axis (UCMP 139803) was recovered, including several cervical vertebrae. These centra possess cervical keels and therefore cannot be referred to *Desmatosuchus* (*Murry & Long, 1989*; *Long & Murry, 1995*) and are assigned to *Calyptosuchus*. The presence of ventral keels on the cervical centra of *Calyptosuchus* is verified by specimen UCMP 27225. *Long & Murry (1995*: fig. 74*)* figured what presumably they thought to represent a cervical series of *Calyptosuchus*, but unfortunately did not provide explicit specimen numbers to identify the specimen further and it could not be located for the current study.

The cervical vertebrae of *Calyptosuchus* are platycoelous, the anterior face being anteriorly concave and the posterior face nearly flat. Both faces are oval and taller than wide. On the anterior cervicals (e.g., UCMP 139793, 139794) the small, subrounded parapophysis is situated at the base of the centrum (Figs. 5F–5I). On more posterior centra (e.g., UCMP 139813) the parapophysis is located closer to the top of the centrum, below the neurocentral suture (Figs. 5J–5M). Anterior cervicals are also

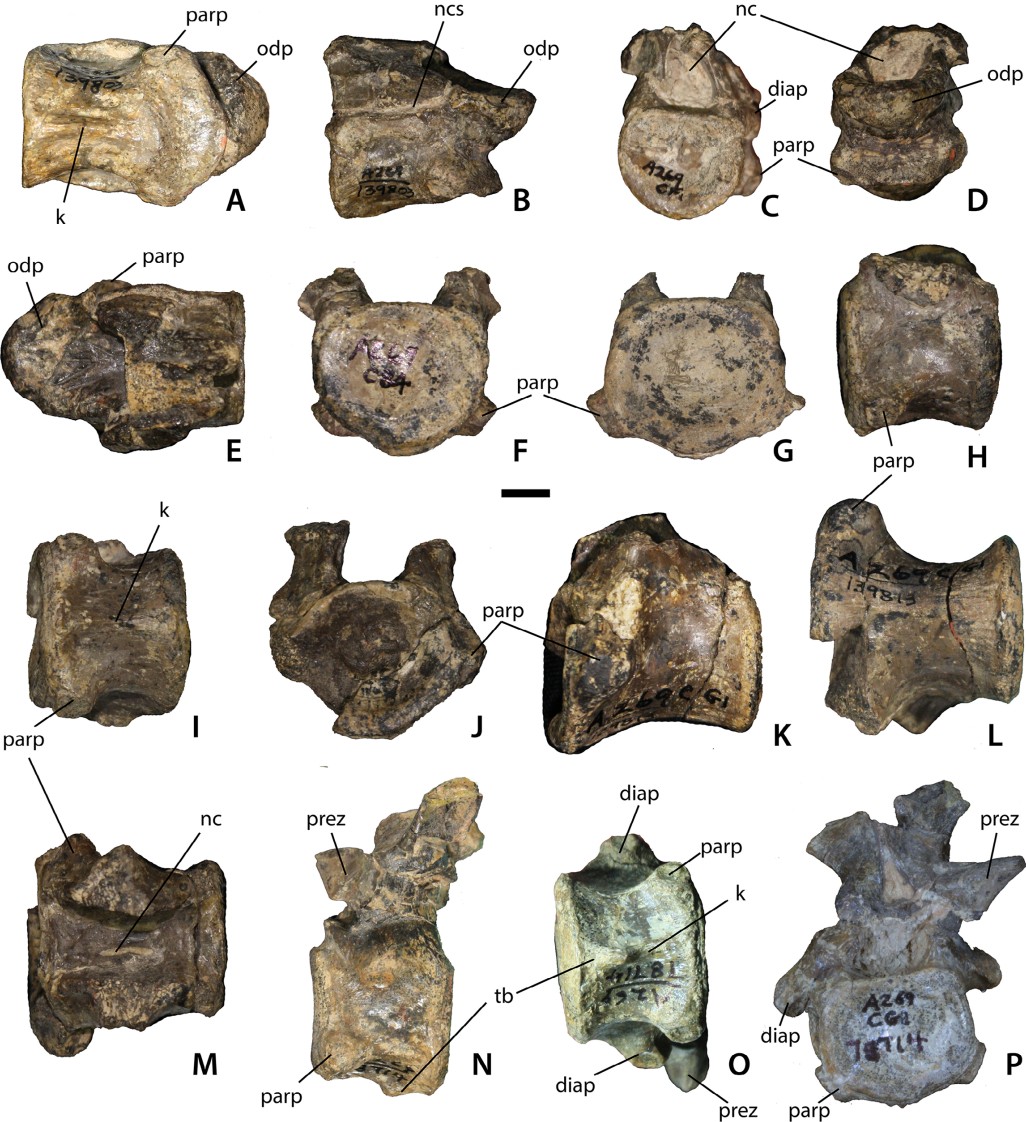

**Figure 5 Cervical vertebrae of *Calyptosuchus wellesi*.** Axial and post-axial cervical vertebrae of *Calyptosuchus wellesi*. (A–E) Axis (UCMP 139803) in ventral (A), lateral (B), posterior (C), anterior (D), and dorsal (E) views, (F), anterior cervical (UCMP 139793) in anterior view, (G), anterior cervical (UCMP 139794) in posterior view, (H–I), anterior cervical (UCMP 139793) in lateral (H) and ventral (I) views, (J–M), posterior cervical (UCMP 139813) in anterior (J), lateral (K), ventral (L), and dorsal (M) views, (N–P), mid-cervical (UCMP 78714) in lateral (N), ventral (O), and anterior (P) views. Scale bar equals 1 cm. diap, diapophysis; k, keel; nc, neural canal; ncs, neurocentral suture; odp, odontoid process; parp, parapophysis; prez, prezygapophyses; tb, ventral tab.

anteroposteriorly shorter than the posterior cervicals (Figs. 5H and 5K). The ventral keel is well-developed and in some specimens (e.g., UCMP 78714) the keel is expanded posteriorly into a small tab (Figs. 5N and 5O). UCMP 78714 also preserves a portion of the neural arch. Although crushed and distorted it shows that the zygapophyses were elongate (Fig. 5P). Prezygadiapophyseal and postzygadiapophyseal laminae (sensu *Wilson, 1999*) are present.

### Trunk vertebrae

The trunk vertebrae of *Calyptosuchus* are more difficult to identify than the cervical vertebrae from the mixed collection of material from the *Placerias* Quarry; however, there are vertebrae with more elongate neural spines that also lack typical accessory articulations (hyposphenes–hypantra) on the neural arch. This readily distinguishes them from the trunk vertebrae of *D. spurensis* which possess much shorter (dorsoventrally) neural spines and exhibit hyposphenes and hypantra (*Parker, 2008b*; *Stefanic, 2017*). The trunk centra of *Calyptosuchus* lack the lateral fossae present in *A. scagliai* (*Desojo & Ezcurra, 2011*). There are also posterior trunk vertebrae preserved in the holotype (UMMP 13950; *Case, 1932*).

UCMP 139694 is most likely the 10th presacral (first trunk) vertebra as it is transitional in the position of the parapophysis between the cervical and trunk series (Figs. 6A and 6B). The parapophysis is situated on the anterodorsal surface of the centrum and it is confluent with the transverse process and connected by a well-developed anterior centrodiapophyseal lamina (acdl; sensu *Wilson, 1999*). In *D. spurensis* this specific placement of the parapopohysis occurs in the 10th presacral position and in the following vertebra (11th presacral) the parapophysis moves onto the transverse process (*Case, 1922*; *Parker, 2008b*). The neural arch of UCMP 139694 also bears a posterior centrodiapophyseal lamina (pcdl) but it is not as well-developed as the anterior centrodiapophyseal lamina (acdl). The joining of these two laminae forms a ventrolaterally opened shallow triangular fossa situated ventral to the transverse process. A postzygadiapophyseal lamina (podl; *Wilson, 1999*) is present as a well-developed thin ridge of bone connecting the transverse process and the postzygapophysis. The centrum is spool-shaped, platycoelous, ventrally smooth, and measures 37.9 mm in length (Fig. 6B). The centrum also has a height of 31.8 mm and a width of 31.6 mm.

UCMP 139796 from CF1 (Figs. 6C–6H) has the typical platycoelous, spool-shape found in aetosaurs and represents a mid-trunk vertebra. The centrum measures 43.4 mm in length, with a height of 35.4 mm and a width of 32.4 mm; thus the lengths of the centra increase along the trunk portion of the vertebral column similar to *D. spurensis* (*Parker, 2008b*). The articular faces of the centrum are nearly flat, but the anterior face is still slightly concave, with expanded rims (Figs. 6C and 6D). The neural arch is taller than the centrum articular faces and the oval neural canal is large (19.4 mm high) (Fig. 6E). In right lateral view the transverse process is mostly broken away (Fig. 6D), but a thick strut originates on the posterolateral corner of the neural arch and terminates on the ventral surface of what is left of the transverse process. This strut represents the posterior centrodiapophyseal lamina (pcdl). A postzygadiapophyseal lamina (podl) forms a shelf from the posterior edge of the transverse process to the right postzygapophysis. A shallow postzygapophyseal centrodiapophyseal fossa (sensu *Wilson et al., 2011*) opens posterolaterally, formed by the junction of these two laminae (Fig. 6D). Although the posterior portion of the neural arch is broken it is clear that there is no deep hyposphene between the postzygapophyses as in *Desmatosuchus* (MNA V9300). The postzygapophyses are not steeply inclined, instead projecting at about 30° above

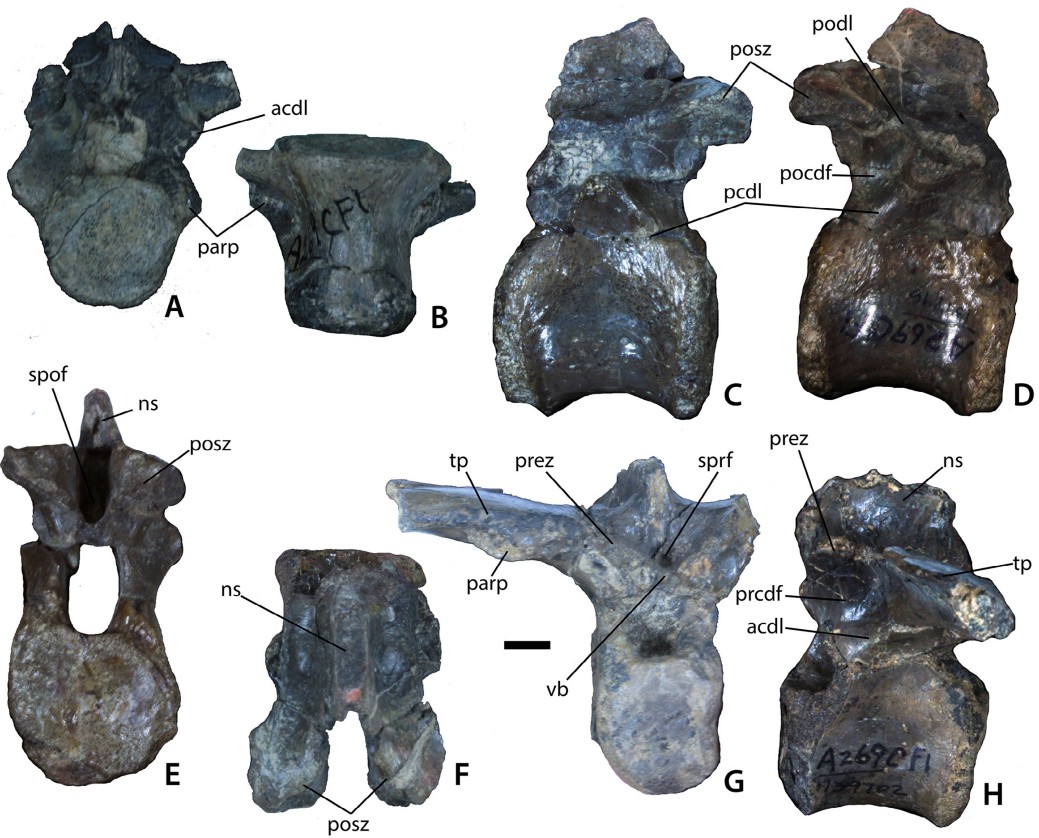

**Figure 6 Trunk vertebrae.** Trunk vertebrae of *Calyptosuchus wellesi*. (A, B) UCMP 139694, 10th presacral vertebra in anterior (A) and ventral (B) views. (C–F) UCMP 139796, mid-trunk vertebra in left lateral (C), right lateral (D), posterior (E), and dorsal (F) views. (G, H) UCMP 139702, posterior trunk vertebra in anterior (G) and lateral (H) views. Scale bar equals 1 cm. acdl, anterior centrodiapophyseal lamina; ns, neural spine; parp, parapophysis; pcdl, posterior centrodiapophyseal lamina; pocdf, postzygapophyseal centrodiapophyseal fossa; podl, postzygapophyseal lamina; posz, postzygapophysis; prcdf, prezygapophyseal centrodiapophyseal fossa; prez, prezygapophysis; spof, spinopostzygapophseal fossa; sprf, spinoprezygapophyseal fossa; tp, transverse process; vb, ventral bar.

horizontal. The postzygapophyses project well posterior to the posterior face of the centrum (Fig. 6F). Anteriorly on the neural arch there is a deep round fossa between the prezygapophyses and the neural spine, the spinoprezygapophyseal fossa (sprf, *Wilson et al., 2011*; Fig. 6D). The neural spine is not anteroposteriorly elongate measuring only about 27 mm at the base and the spinal laminae are present but weakly developed.

Another trunk vertebra from CF1 (UCMP 139702) preserves a few more details. In front of the anterior fossa (sprf) described for UCMP 139796, the prezygapophyses meet to form a broad shelf or ventral bar (Fig. 6G) as in *S. robertsoni* (*Walker, 1961*: fig. 7j). There is no hypantrum. The right transverse process is nearly complete. It is broad, about 26.7 mm in width, compared to the centrum, which has a width of 25.7 mm. The upper surface of the transverse process is flat and the ventral surface thickened with the strut described for UCMP 139796, which continues onto the base of the neural arch. The

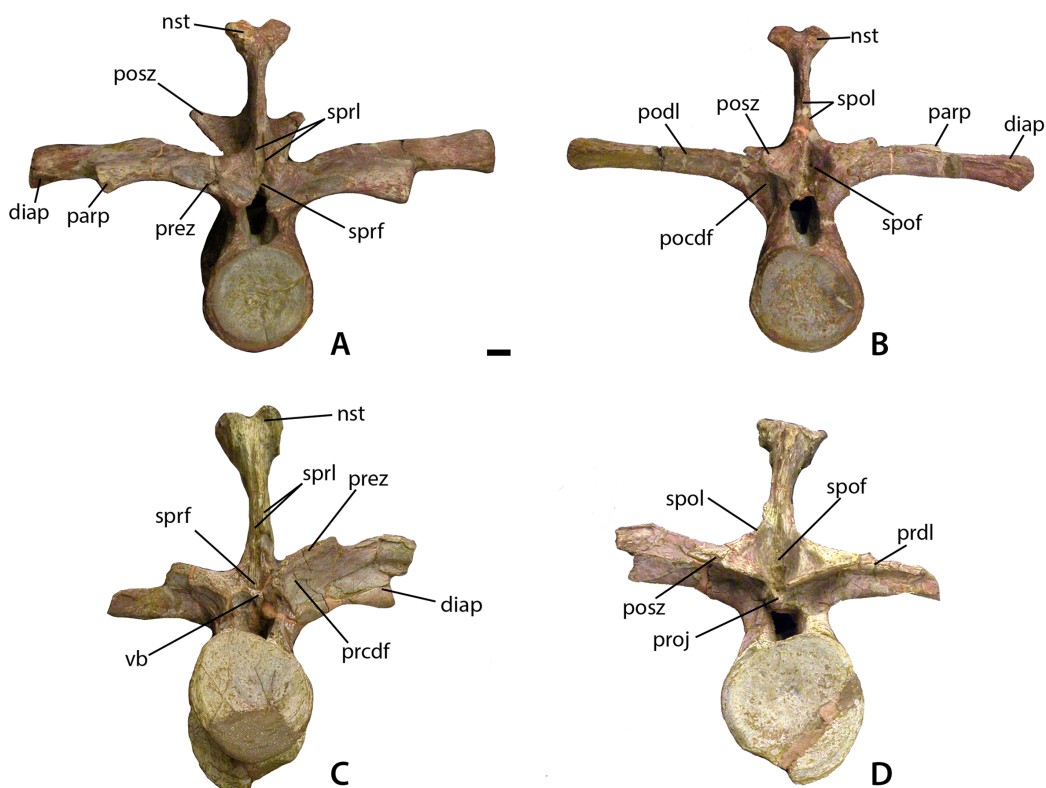

**Figure 7 Mid-trunk vertebrae of *Calyptosuchus wellesi*.** Mid-trunk vertebrae of *Calyptosuchus wellesi* (UMMP 7470). (A, B) Vertebra in anterior (A) and posterior (B) views. (C, D) Vertebra in anterior (C) and posterior (D) views. Scale bar equals 1 cm. nst, neural spine table; parp, parapophysis; prdl, prezygapophyseal lamina; pocdf, postzygapophyseal centrodiapophyseal fossa; podl, postzygapophyseal lamina; posz, postzygapophysis; prcdf, prezygapophyseal centrodiapophyseal fossa; prez, prezygapophysis; proj, posterior projection; spof, spinopostzygapophseal fossa; spol, spinopostzygapophyseal lamina; sprf, spinoprezygapophyseal fossa; sprl, spinoprezygapophyseal lamina; vb, ventral bar.

parapophysis is positioned 29.3 mm laterally from the origin of the transverse process. The distal end of the transverse process, the diapophysis, is not preserved but even incomplete, the process has a length of 44.4 mm. The zygapophyses are inclined at close to 45° to the horizontal. The centrum length is 39.9 mm long and 28.3 mm high.

A third trunk vertebra from CF1 (UCMP 139795) preserves the postzygapophyseal region extremely well. As with the other trunk vertebrae there are no accessory processes (hyposphene). Instead at the base of the medial union of the postzygapophyses there is a small posteriorly pointed projection that would rest on top of the ventral bar formed by the joined prezygapophyses of the subsequent vertebra. This pointed projection also occurs in *S. deltatylus* (PEFO 34045). The ventral bar and posterior projection in the trunk vertebrae is also shared with some phytosaurians (e.g., *Smilosuchus*, TMM 43685-206).

Two other well-preserved trunk vertebrae (Figs. 7A–7D) referable to *C. wellesi* are from UMMP 7470, which includes a partial sacrum and the two trunk vertebrae, as well as two paramedian osteoderms. The best preserved vertebra is a nearly complete anterior mid-trunk vertebra (*Case, 1932*: figs. 2–4). The centrum is laterally compressed and

ventrally concave because of the flaring articular rims. It has a length of 48.7 mm, and width of 42.3 mm, and a height of 42.8 mm. The neural arch and spine are tall, twice the height of the centrum at 78.8 mm, with 55.2 mm for the neural spine height. The neural spine is mediolaterally thin, expanded anteroposteriorly (34.2 mm long) and terminates with a pronounced lateral expansion (spine table). The postzygapophyses extend posteriorly beyond the posterior articular face of the centrum and are oriented at 45° above horizontal. The prezygapophyses form a flat plate almost indistinguishable from the transverse processes (Figs. 7A and 7C). The transverse processes are broad with a flat dorsal surface, and nearly twice the width of the centrum (82.3 mm). The processes are of the typical aetosaurian arrangement with both rib articulations situated on the transverse processes (Figs. 7A and 7C). Transverse processes and postzygapophyses are connected by a thin sharp postzygapophyseal lamina (podl), which forms the deep spinopostzygapophyseal fossa (spof) just anterior to the postzygapophyses (Figs. 7B and 7D).

*Long & Murry (1995)*: fig. 75a) considered the transverse processes of the dorsal series extremely elongate throughout the entire column. However, they figured posterior trunk vertebrae of UMMP 13950 as an example, which have the ribs fused to the transverse processes, giving the appearance of greatly elongate processes (as noted by *Case, 1932*). This fusion of transverse process and rib is also found in *S. deltatylus* (PEFO 34045) as well as *D. spurensis* (MNA V9300; *Parker, 2008b*). However, the processes in *C. wellesi* differ from those two taxa in that they are flat dorsoventrally and anteroposteriorly broad (*Case, 1932*: pl. 4, fig. 1). The centra of the posterior most trunk vertebrae are anteroposteriorly short in comparison with those of the mid-trunk vertebrae, with large flaring articular rims.

### Sacral vertebrae

The best preserved sacral vertebrae are in the holotype (UMMP 13950) as well as in the partial pelvis (UMMP 7470) and were well-described and figured by *Case (1922, 1929, 1932)*. There are two vertebrae in the series, which differ from those of desmatosuchine aetosaurs in that they are not fused to each other (*Parker, 2008b*; *Griffin et al., 2017*) although *Case (1932)* noted that the zygapophyses between the two sacral vertebrae were reduced in size. The articular faces of the centra are round. The neural arches are robust and bear the heavy, expanded sacral ribs, and the neural spines are also robust and taller than the centra. The neural spines possess expanded apices or "spine tables."

An isolated specimen (UCMP 139785) from grid block C78W in the *Placerias* Quarry is most likely referable to *C. wellesi* as it does not show fusion to the other sacral as do others in the collection (e.g., UCMP 139787). The vertebra is very massive with the proximal portions of the sacral ribs firmly sutured to the neural arch (Figs. 8A–8D). The upper surface of the ribs is swept posteriorly (Fig. 8B). The centrum faces are roughly "heart-shaped" and the ventral surface lacks a keel (Figs. 8C and 8D). The neural spine is broken off, but was obviously robust (thick and elongate) as in UMMP 7470. There is a distinct spinoprezygapophyseal fossa (Fig. 7A) under the prezygapophyses.

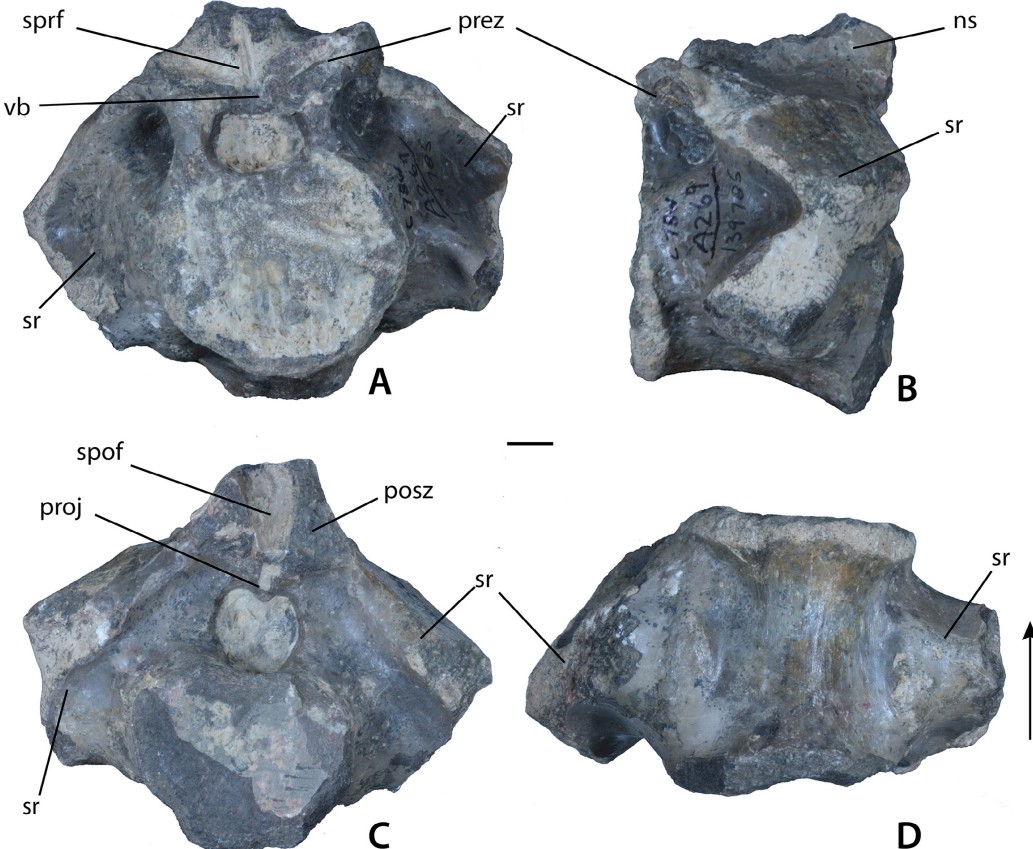

**Figure 8 Sacral vertebrae of *Calyptosuchus wellesi*.** (A–D) Sacral vertebra of *Calyptosuchus wellesi* (UCMP 139785) in anterior (A), lateral (B), posterior (C), and ventral (D) views. Scale bar equals 1 cm. ns, neural spine; posz, postzygapophysis; prez, prezygapophysis; proj, posterior projection; spof, spinopostzygapophseal fossa; sprf, spinoprezygapophyseal fossa; sr, sacral rib; vb, ventral bar.

### Caudal vertebrae

The *Placerias* Quarry collection contains dozens of aetosaur caudal centra with broken neural arches; however, at this time it is not possible to assign these elements to particular taxa. However, the first seventeen vertebrae of the caudal series of *C. wellesi* are well-preserved in articulation in the holotype (UCMP 13950) and were described by *Case (1932)*. The most notable feature of the caudal series of UCMP 13950 is the height of the neural spines, which is greater than the height of the centrum. This differs from aetosaurs such as *D. spurensis* (MNA V9300) and *Paratypothorax* (PEFO 3004) where the height of the neural spine is equal to or less than the height of the centrum. It is similar to the condition in *A. scagliai* (PVL 2073) and *S. robertsoni* (*Walker, 1961*: fig. 10).

*Long & Murry (1995*: 83) state that the ventral grooves of the caudal centra in *C. wellesi* are narrower than those of *D. spurensis* and "bear faint, longitudinal ridges." However, they provide no basis for their taxonomic referrals nor any specimen numbers, so this claim cannot be verified. The caudal ribs or transverse processes of paratypothoracins

originate close to the base of the centrum (e.g., PEFO 3004). No centra with low caudal ribs are currently known from the *Placerias* Quarry, and thus all of the preserved centra presumably belong to *C. wellesi* or *D. spurensis* although they cannot be distinguished between those taxa.

### Scapulocoracoid

No bones of the pectoral girdle are preserved in the holotype of *C. wellesi* (UMMP 13950). *Long & Murry (1995)* assign several scapulocoracoids (UCMP 78698, UCMP 32196, UCMP 27976) from the *Placerias* Quarry to *C. wellesi*; however these elements were recovered from areas CD and CE which provided many osteoderms of *D. spurensis* and none referable to *C. wellesi* (Fig. 2). Furthermore, coracoids assigned to *C. wellesi* (UCMP 32196, UCMP 27976; *Long & Murry, 1995*) are from C8 and C75W, and also from areas that provided predominantly material of *Desmatosuchus* (Fig. 2). Thus, none of the *Placerias* Quarry material can be unambiguously assigned to *C. wellesi*. Differences between the coracoids of *D. smalli* (TTU P-9023) and *S. robertsoni* (*Walker, 1961*) pertain to the development of the subglenoid buttress. Unfortunately this area is not preserved in any of the *Placerias* Quarry specimens.

### Forelimb

As with the shoulder girdle, no forelimb elements are present in the holotype of *C. wellesi* (UCMP 13950). Moreover, *Long & Murry (1995)* did not assign any forelimb material to *C. wellesi*. The UCMP *Placerias* Quarry collection contains numerous aetosaur humeri, but none can be clearly referred to *C. wellesi*.

### Pelvic girdle

Several pelvic girdles have been referred to *C. wellesi* including the holotype (UCMP 13950; Fig. 9), a specimen from the Dockum Group of Texas (UMMP 7470), and elements from the *Placerias* Quarry (*Case, 1929*, *1932*; *Long & Murry, 1995*). The *Placerias* Quarry elements include a left ilium (UCMP 32422) and a corresponding left ischium (UCMP 32148), both from grid CF1 (Figs. 10A and 10B), and figured by *Long & Murry (1995*: figs. 79–80*)*. The collection from CF1 also contains a crushed, but complete right ilium (UCMP 25941) and a right ischium (UCMP 32153) (Fig. 10C). These elements match the two figured by *Long & Murry (1995)* perfectly and all four elements probably belong to the same individual (*Long & Murry, 1995*). The difference in color between these elements in Fig. 10 is a photographic lighting artifact. Grid CF1 contains a fair amount of material referable to *Calyptosuchus*, mainly cervical vertebrae, including some paramedian osteoderms, so referral of these pelvic elements to *C. wellesi* is supported.

The problem with assigning isolated ilia from the quarry to specific taxa is that the morphology of the ilium of *Desmatosuchus* is poorly understood. The holotype of *D. spurensis* (UMMP 7476) preserves only a fragmentary left ilium that is missing almost the entire posterior portion of the iliac blade. A referred specimen of *D. spurensis* (MNA V9300) as well a specimen of *D. smalli* (TTU P-9172) preserve nearly complete sacra; however, the anatomy of the ilia is difficult to interpret on these specimens because they are highly distorted, in part because of the complete fusion of the sacral ribs to the ilia

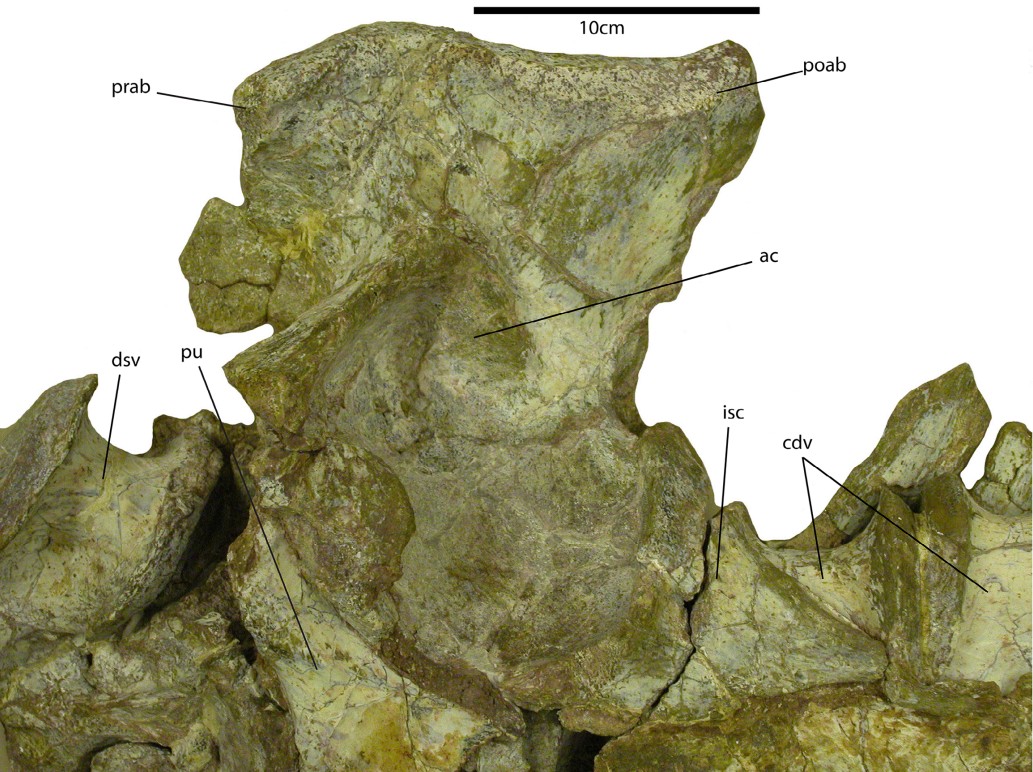

**Figure 9 Sacrum of *Calyptosuchus wellesi*.** Portion of the sacrum and vertebral column of the holotype specimen of *Calyptosuchus wellesi* (UMMP 13950) in ventral view. ac, acetabulum; cdv; anterior caudal vertebra; dsv, posterior trunk vertebra; isc, left ischium; poab, postacetabular blade of the left ilium; prab, preacetabular blade of the left ilium; pu, left pubis.

(see *Parker, 2008b*). *Long & Murry (1995*: figs. 91–92*)* assigned an isolated right ilium from Crosby County Texas (UMMP 7322) to *D. spurensis*. This specimen possesses an acute angle between the anterior portion of the iliac blade and the anterior edge of the iliac body as well as a triangular (in lateral view) posterior iliac blade. The holotype ilium (UMMP 7476) as preserved is consistent with this although much of the anterior portion of the iliac blade is damaged. If UMMP 7322 is indeed referable to *D. spurensis* UCMP 32422 differs from it mainly in that the posterior iliac blade is squared off and not pointed as in UMMP 7322. This is the character *Long & Murry (1995)* used to assign ilia to *C. wellesi* and this referral is followed here.

### Ilium

The ilia in *C. wellesi* have ventrally directed acetabula; however, to make the following description easier to follow the element is described as if it is oriented vertically, thus the iliac blade is dorsal and the acetabulum ventral and lateral. The preacetabular process of the iliac blade in UCMP 25941 is short and does not extend far anterior of the pubic peduncle (Figs. 10A and 10B). It is 50 mm long, mediolaterally thick and triangular in lateral view with a ventrally curved tip. The postacetabular portion of the iliac blade extends well beyond the posterior edge of the pubic peduncle and is thickened very close

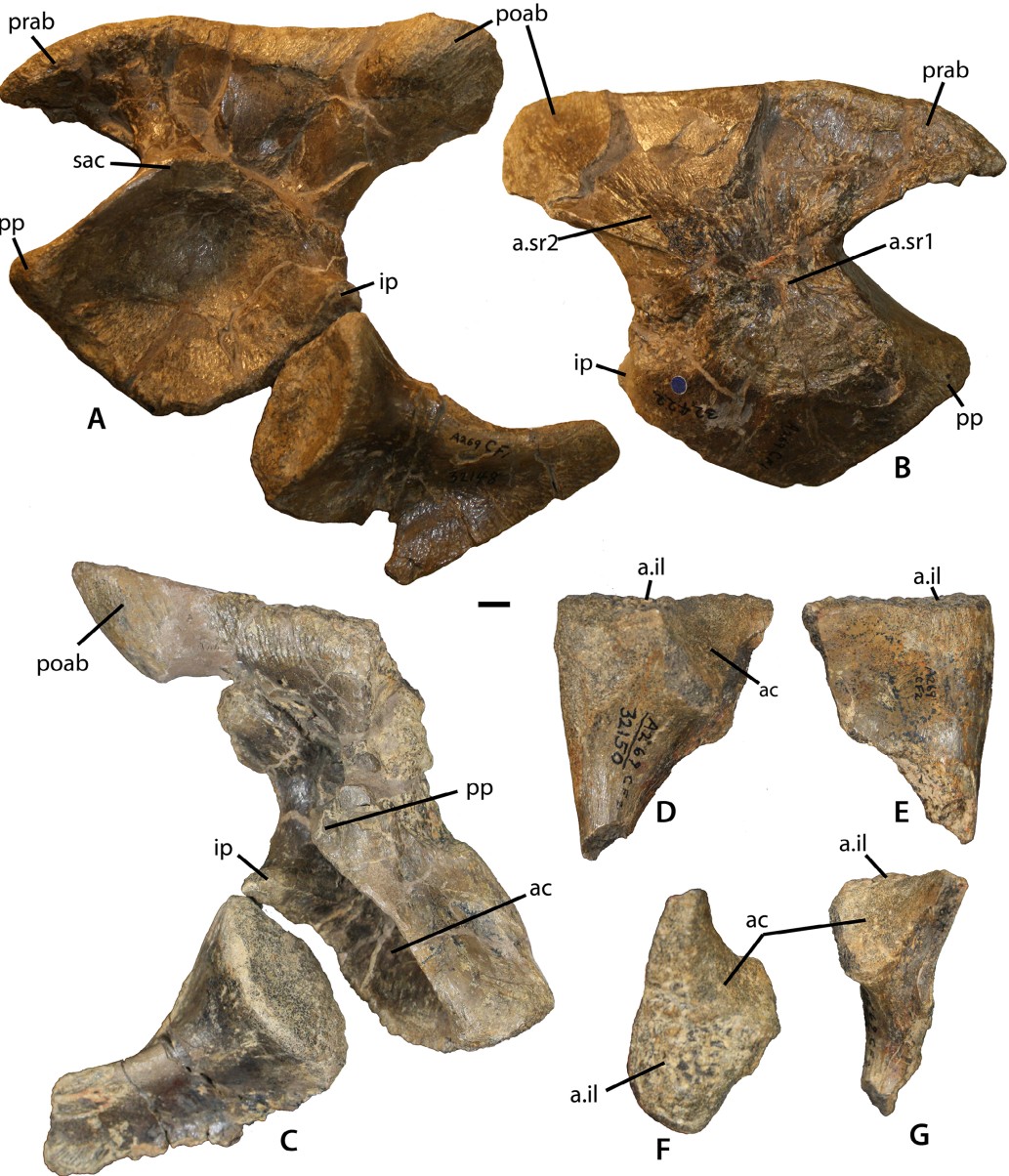

**Figure 10 Referred sacrum from the *Placerias* Quarry.** Pelvic elements of *Calyptosuchus wellesi*, possibly from a single individual. (A) Left ilium (UCMP 25941) and ischium (UCMP 32148) in lateral view (see text about anatomic directions for the pelvic elements). (B) Left ilium (UCMP 25941) in medial view. (C) Right ilium (UCMP 25941) and ischium (UCMP 32153) in lateral view. (D–G) left pubis (UCMP 32150) in lateral (D), medial (E), dorsal (F), and posterior (G) views. Scale bar equals 1 cm. ac, acetabulum; a.il, articulation with ilium; a.sr1, articulation with sacral rib1; a.sr2, articulation with sacral rib2; il, ilium; ip, ischiadic peduncle; poab, postacetabular blade; pp, public peduncle; prab, preacetabular blade; sac, supraacetabular crest.

to its proximal end. The entire iliac blade is 180 mm long, and 52 mm high above the acetabulum. The dorsal surface is highly rugose, marked with scars for the attachment of the *M. iliotibialis* 1–3 (*Schachner, Manning & Dodson, 2011*). The acetabular area is roughly diamond-shaped in lateral view and delineated dorsally by a well-developed supraacetabular rim (Fig. 10A). The main iliac body is slightly concave dorsal to the

acetabulum, lacking the deep recess found between the supraacetabular rim and the posterior portion of the iliac blade in *S. deltatylus*.

The pubic and iliac peduncles are thickened anteriorly and posteriorly respectively, and both are comma-shaped in ventral views. The two peduncles meet at a ventrally directed point ventral to the iliac portion of the acetabulum. Medially, there are scars for the two sacral ribs, which cover not only the iliac neck but also a large portion of the ilium ventral to the iliac blade and medial to the acetabulum (Fig. 10B). This is a result of the ventrally directed acetabula as in *A. scagliai* (PVL 2073) and *Typothorax coccinarum* (PEFO 33967). The iliac blade thins dorsally from the sacral rib scars. Overall the ilium of *C. wellesi* is very similar to that of *A. scagliai* (PVL 2073) and *Ebrachosaurus singularis* (*Kuhn, 1936*). It differs from *N. engaeus* (PVL 3525) in having a much more robust anterior process of the iliac blade. It differs significantly from the ilium of *T. coccinarum* (UCMP 122683) which has a taller, but anteroposteriorly shorter iliac blade, as well as a more gracile, and "hooked" anterior process which does not extend anteriorly beyond the pubic peduncle (*Long & Murry, 1995*: figs. 106–107). The right ilium is well-preserved in the referred specimen UMMP 7470 (*Case, 1922*: fig. 28b). It is nearly identical to UCMP 25941 with the thickened, short, recurved anterior iliac blade. Both ilia are present in the holotype (UMMP 13950) but both are incomplete, crushed, and presently badly broken (Fig. 9; *Case, 1932*, pl. II). Note that the photo of the pelvic girdle and vertebral column in Plate II in *Case (1932)* is reversed.

### Ischium

The left ischium (UCMP 32148) associated with the UCMP ilium described above is nearly complete (Fig. 10A). It is anteroposteriorly short, not much longer than tall, with a length of 110 mm and a height of 97 mm. This differs from the ischia of *A. scagliai* (PVL 2073), *S. robertsoni* (*Walker, 1961*), and *Aetosaurus ferratus* (*Schoch, 2007*), where the posterior process is more elongate. The pubic peduncle is comma-shaped in dorsal view and contacts the corresponding peduncle of the ilium. The oval acetabular surface is deeply concave and bordered posteriorly and ventrally by a strongly raised, curved rim. The main body of the ischium is essentially a thickened "rod" that curves posteriorly and dorsally. A mediolaterally thin flange of bone extends ventrally for the entire length of the "rod" (Fig. 10A). The ventral margin is straight. The lateral surface of the thin flange is rugose presumably for attachment of the third head of the *M. puboischiofemoralis externus* (*Schachner, Manning & Dodson, 2011*). Medially there is an elongate suture for the opposing ischium. The anterior margin bears a distinct notch. This notch is also present on the right ischium of UMMP 7470. The posterior process of UMMP 7470 is more elongate than that of UCMP 32148, but still not as elongate as in *Walker's (1961)* reconstruction of *S. robertsoni*. The ischia are also present in UMMP 13950 but are poorly preserved (Fig. 9). *Case (1932*: pl. III) restores the ischium as dorsoventrally deep and anteroposteriorly short, consistent with UCMP 32148.

### Pubis

The best preserved pubis from the *Placerias* Quarry material is a left element (UCMP 32150) from grid CF2 (Figs. 10D–10G). It shares the same preservation, color, and size

with the ilium and ischium described above, but does not quite articulate and thus may not belong to the same individual. The pubic rod is slender and its distal end is broken away (Figs. 10D and 10E). The concave acetabular surface is reduced compared to the area on the ischium and there is a groove just ventral to this surface. The articular surface for the ilium is comma-shaped in dorsal view (Fig. 10F). The obturator flange is broken away (Fig. 10G) so the number of openings in this element cannot be determined. *Walker (1961)* restored the pubis of *S. robertsoni* with pubic foraminae and a pubis of *S. deltatylus* (PEFO 31217) also has two openings. Only a single foramina is present in the pubis of *D. spurensis* (MNA V9300) and the number of foraminae is unknown in *A. ferratus* (*Schoch, 2007*).

The proximal portion of the right pubis is present in UMMP 7470 (*Case, 1922*: fig. 28b). The posterior margin as preserved shows the anterior border of an obturator foramen but the element is not complete enough to determine if there was a second opening. The proximal head of UMMP 7470 bears a deep lateral groove that originates at the acetabular rim and extends parallel to the anterior margin of the pubis. The distal end of the element is broken away so that the extent of the groove cannot be determined. This groove is only weakly developed in UCMP 32150, which is also missing its distal end. UMMP 13950 preserves the distal end of the pubis, which expands into the broad pubic "apron" typical for suchians (*Case, 1932*). *Case (1932*: pl. III*)* reconstructs the girdle as dorsoventrally shallow with the distal margin of the pubis at the same horizontal level as the ventral margin of the ischium. This differs greatly from the condition in *D. spurensis* (MNA V9300) where the pubis extends well below the level of the ischium, but is similar to the short pubes of *T. coccinarum* (*Long & Murry, 1995*).

The distal end of the pubic rod extends slightly beyond the ventral margin of the pubic apron, as is typical for aetosaurs. This end is slightly swollen as in *S. robertsoni* (*Walker, 1961*), but does not form the distinct knobby pubic boot found in *D. spurensis* (MNA V9300).

### Femur

The best preserved femur that can be referred to *C. wellesi* is UCMP 25918, which is a left side element from CF1 (Figs. 11A–11D; *Long & Murry, 1995*: figs. 81, 83). It is of similar preservation and the right size to match the pelvic elements described above so it is very possible that all of these elements belong to a single individual. *Long & Murry (1995)* describe it as "more gracile" than femora from the quarry that they assign to *D. spurensis*. Overall it is less sigmoidal than the femur of phytosaurs, as is characteristic of aetosaurs (Figs. 11A–11C). It has a total length of 329 mm. The proximal head is badly eroded (Figs. 11A and 11B). The fourth trochanter is a pronounced crescent-shaped ridge located about 120 mm ventral to the proximal end (Fig. 11A). The distal femoral condyles are well-preserved (Fig. 11D). The medial condyle has a posteromedial corner with an angle of 90° and a rounded anteromedial corner. The lateral condyle is larger than the medial and anterolaterally bears a distinct crista tibiofibularis. The angle between the crista tibiofibularis and the lateral condyle is obtuse. The posterolateral corner of the lateral condyle is rounded and expanded posteriorly.

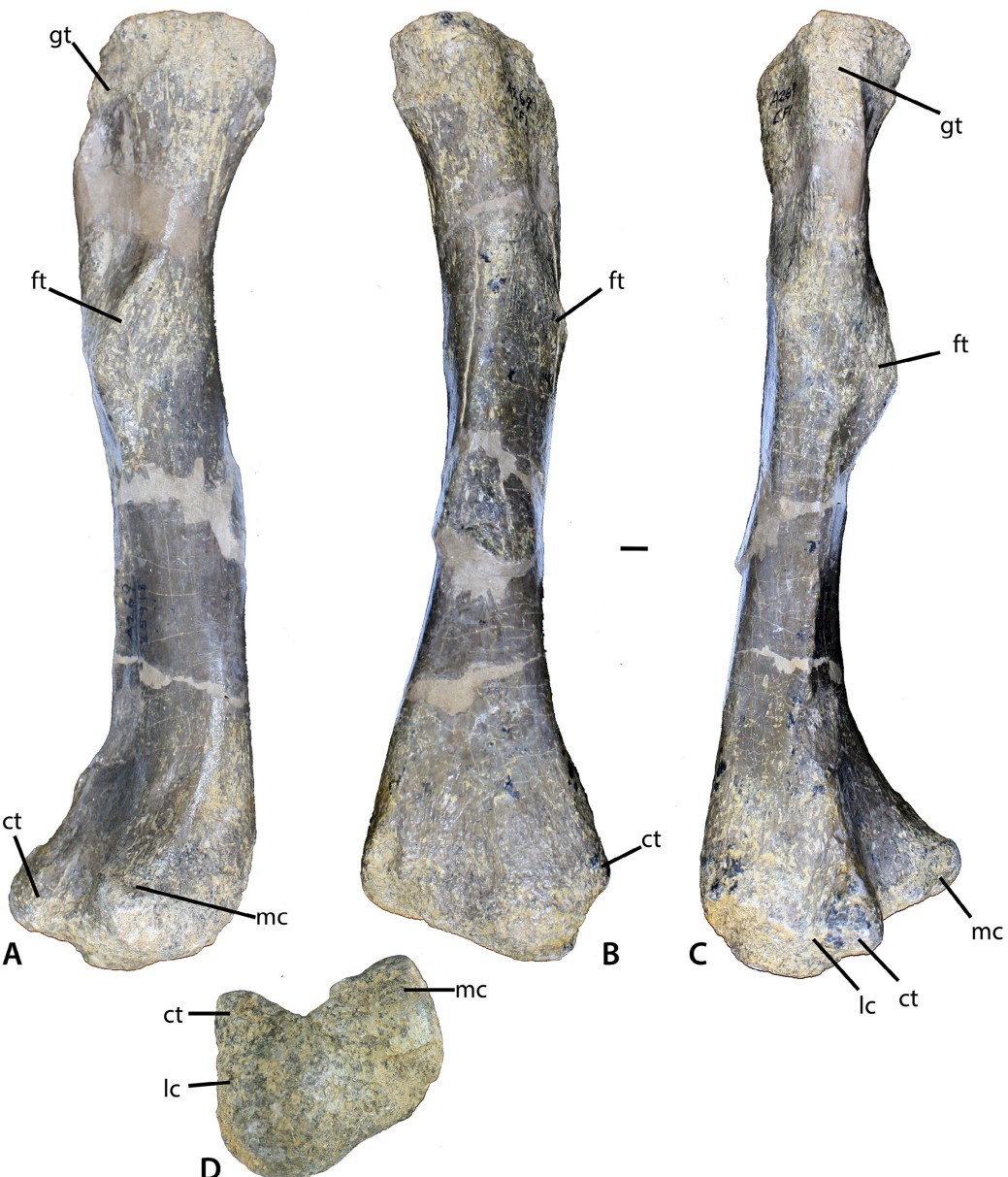

**Figure 11 Femur of *Calyptosuchus wellesi*.** (A–D) Left femur of *Calyptosuchus wellesi* (UCMP 25918) in posteromedial (A), medial (B), lateral (C), and distal (D) views. Scale bar equals 1 cm. ct, crista tibiofibularis; ft, fourth trochanter; gt, greater trochanter; lc, lateral condyle; mc, medial condyle.

## *Tibia*

UCMP 25887 from C64M occurs within a cluster of osteoderms of *C. wellesi*, but material referable to *D. spurensis* occurs in that grid as well. Nonetheless, this left tibia is much more gracile than others found in the quarry (e.g., UCMP 25877), which probably belong to *Desmatosuchus* (Fig. 12; *Long & Murry, 1995*). UCMP 25887 (Figs. 13A–13D) has a length of 186 mm, shorter than the femur as is typical for aetosaurs. The proximal head is oval in proximal view with a width of 73 mm, a length of 52 mm and

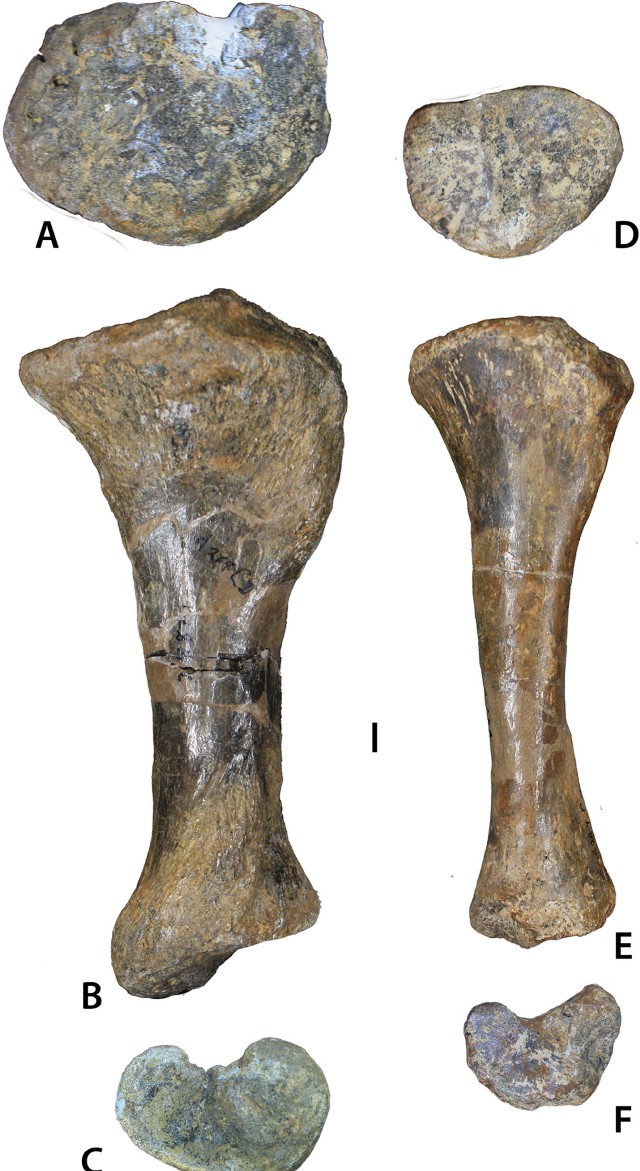

**Figure 12 Aetosaurian tibiae from the *Placerias* Quarry.** (A–C) *Desmatosuchus spurensis* left tibia (UCMP 25877) in proximal (A), posterior (B), and distal (C) views. (D–F) *Calyptosuchus wellesi* left tibia (UCMP 25887) in proximal (D), posterior (E), and distal (F) views. Scale bar equals 1 cm.

is divided into two distinct sections by a nearly central ridge. The medial surface has slightly more area than the lateral surface and it is concave, whereas the lateral surface is convex. A cnemial crest is absent (*Nesbitt, 2011*), and there is a distinct "lip" posteriorly on the lateral portion of the head. The posterior portion of the distal end possesses a dorsoventrally oriented groove (*Nesbitt, 2011*: char. 337–1) for articulation with the astragalus. There is some damage to the medial condyle of the distal end in UCMP 25887. Overall there are few noticeable differences in the distal ends of UCMP 25887 and UCMP 25877 other than size. However, the proximal end in UCMP 25877 is much more

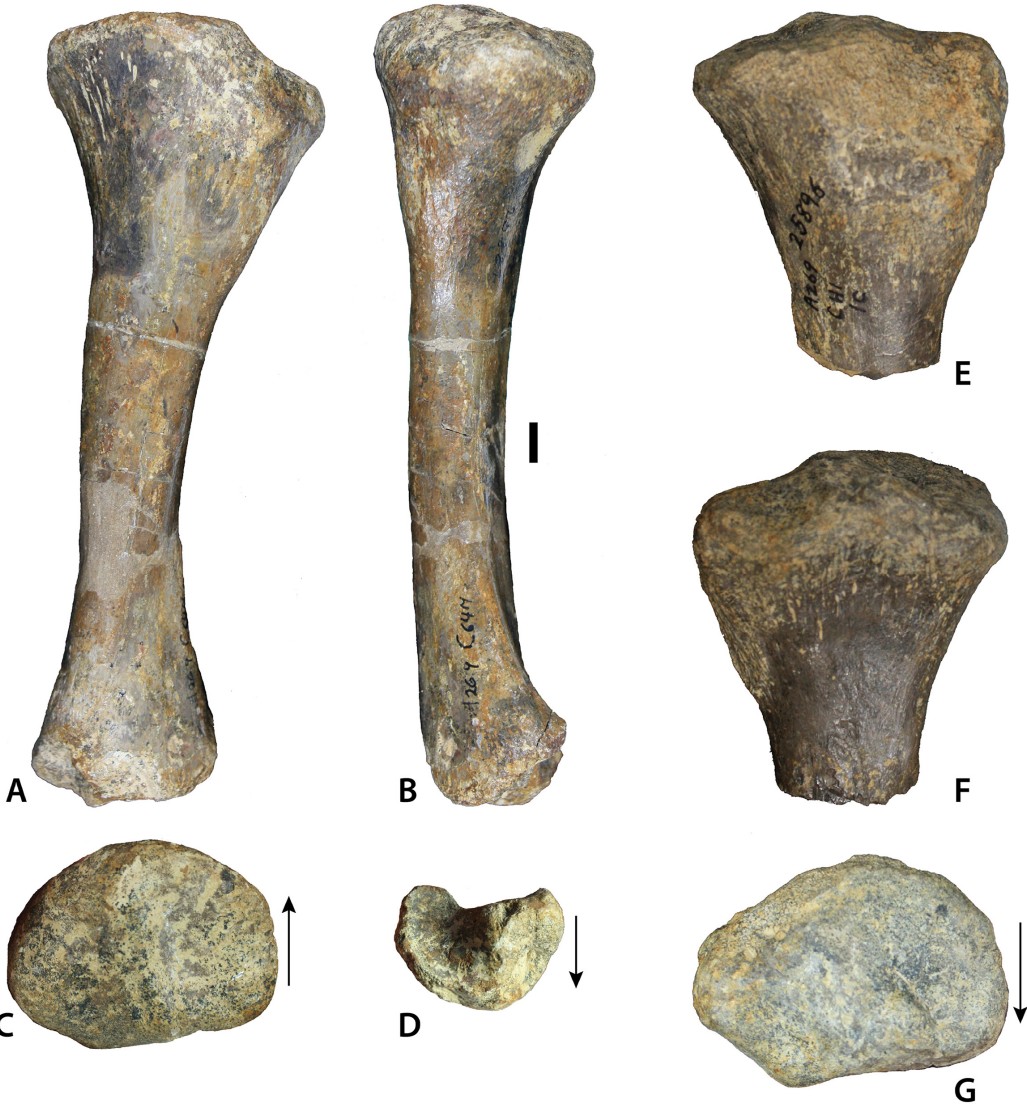

**Figure 13 Tibia of *Calyptosuchus wellesi*.** Tibiae of *Calyptosuchus wellesi*. (A–D) UCMP 25887, left tibia in posterior (A), medial (B), proximal (C), and distal (D). (E–G) UCMP 25896, proximal end of left tibia in posterior (E), anterior (F), and proximal (G) views. Scale bar equals 1 cm. Arrows indicate anterior direction.

expanded medially and has a distinct dorsal notch on the dorsolateral surface. There are two other gracile tibiae in the *Placerias* Quarry collection; UCMP 25896 (Figs. 13E–13G) is a left tibia from grid CH1, and UCMP 25894 is a left tibia from grid CH2 that was figured by *Long & Murry (1995*: fig. 84*)*.

### Fibula

UCMP 25802 from grid C67M is gracile compared to other fibulae in the *Placerias* Quarry collection and, as preserved, matches much of the material of *C. wellesi*. *Long & Murry (1995)* also assigned this element to *C. wellesi*. The specimen represents the proximal end of a left fibula. The iliofibularis trochanter is broken off. There is a small tubercle on the

medial side of the shaft. *Long & Murry (1995*: 84*)* state that "the diagonal ridge, so prominently exhibited along the medial fibular shaft of *Desmatosuchus* [*spurensis*], may not have been present in [*Calyptosuchus*] *wellesi*." However, UCMP 25802 is not complete enough to evaluate this claim.

### Astragalus

There are many astragali in the *Placerias* Quarry collection, but none fits the gracile tibiae in the collection that probably represent *C. wellesi*. *Long & Murry (1995)* figured and assigned a right astragalus from grid CF2 to *C. wellesi* (UCMP 34485); however, this specimen is currently on loan to another researcher and I was unable to examine it. Nonetheless, *Long & Murry (1995)* stated that they were unable to differentiate between the astragali of *Desmatosuchus* and *Calyptosuchus* and thus it is unclear how this assignment was originally made. Thus, neither the type nor referred specimens of *C. wellesi* preserve the astragalus.

### Calcaneum

As with the astragali there are lots of aetosaur calcanea in the collections as well, but as the calcaneum of *Desmatosuchus* is unknown, they cannot be differentiated. *Long & Murry (1995*: fig. 82*)* figured a left calcaneum (UCMP 34481) from CG1 as pertaining to *C. wellesi*. It is not clear what characters they used to make this assignment. UCMP 34481 is very similar to the calcaneum of *A. scagliai* (PVL 2073) with a dorsoventrally flattened, mediolaterally expanded posterior tuber, and a deep concavity on the ventral surface of the anterior portion of the tuber. This deep concavity is sharply rimmed and also prominent in *T. coccinarum* (AMNH FR 2713).

### Osteoderms

The holotype of *C. wellesi* (UMMP 13950) preserves an articulated set of osteoderms starting with the posterior dorsal trunk series and extending back through much of the tail (Fig. 14). These include trunk, lateral, and appendicular osteoderms and, importantly, they are associated with a vertebral column to aid with placement of specific rows. A significant landmark is the neural spine pushed up through the dorsal carapace, which is that of the first caudal vertebra (*Case, 1929*). Accordingly I have placed it between the first and second caudal paramedians where it pushed the first paramedian anteriorly and displaced the second paramedian posteriorly (Figs. 14 and 15). UMMP 13950 was thoroughly described by *Case (1932)* and is not in need of a full redescription.

Referred specimens from the St. Johns, Arizona area (Blue Hills, *Placerias* Quarry) provide more details regarding the mid-dorsal region as well as the ventral trunk osteoderms. Cervical osteoderms are currently unknown for *C. wellesi*. The cervical lateral plates assigned by *Long & Ballew (1985)* to *C. wellesi* that were reportedly characteristic of the genus (*Long & Murry, 1995*) actually belong to a paratypothoracin aetosaur, most likely *Tecovasuchus* (*Parker, 2005*; *Heckert et al., 2007*).

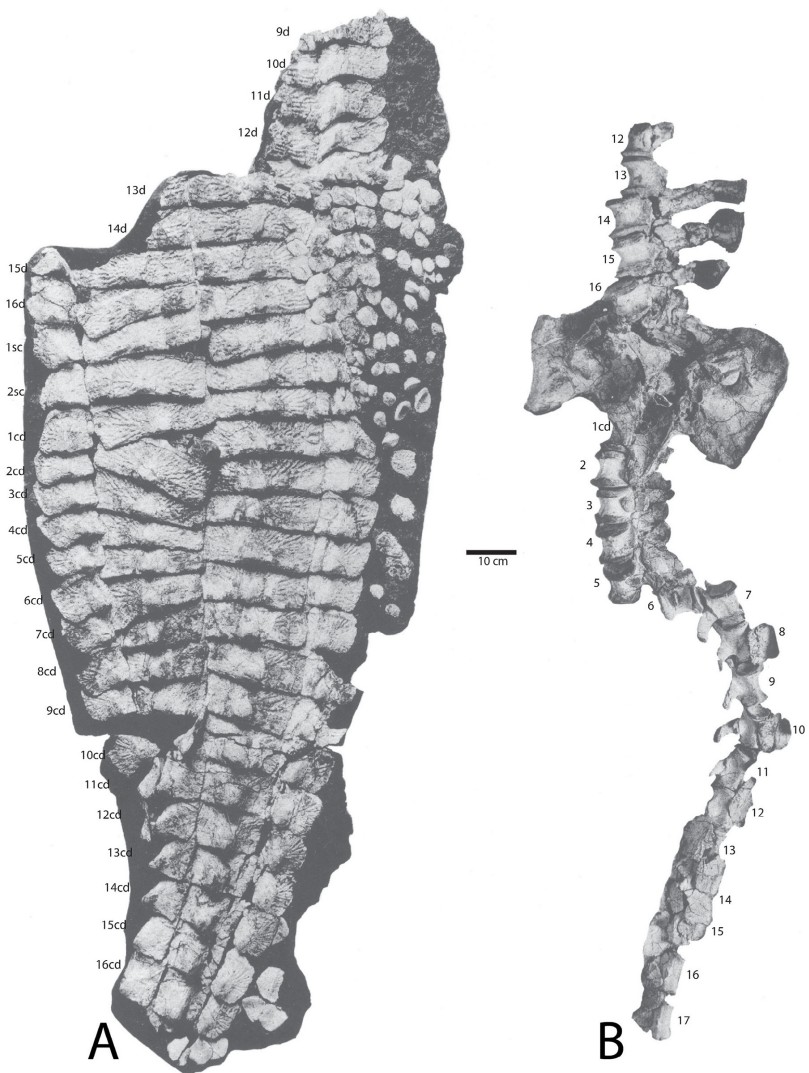

**Figure 14 Holotype specimen of *Calyptosuchus wellesi* (UMMP 13950).** Holotype specimen of *Calyptosuchus wellesi* (UMMP 13950) showing assigned positions of (A) osteoderms, (B) pelvis, and vertebral column. Modified from *Case (1932)*. d, trunk position; sc, sacral position; cd, caudal position.

### Paramedian osteoderms

*Trunk series*

The holotype of *C. wellesi* (UMMP 13950) preserves the last four presacral paramedians of the right side and the last two of the left side as well as the two sets that would have been situated over the sacrum (Figs. 14 and 15). The osteoderms bear strongly raised anterior bars with anterolateral projections, sigmoidal lateral and straight medial margins. The dorsal eminence is a broad, low pyramidal structure that contacts and slightly overhangs the posterior plate margin. The boss is slightly situated medially on the osteoderm surface. A strongly developed pattern of pits and elongate grooves and ridges radiates from the position of the eminence. This ornamentation strongly differs from that of *S. robertsoni* (NHMUK 4789a) and *S. olenkae* (ZPAL AbIII 570/1) where the radiating grooves and

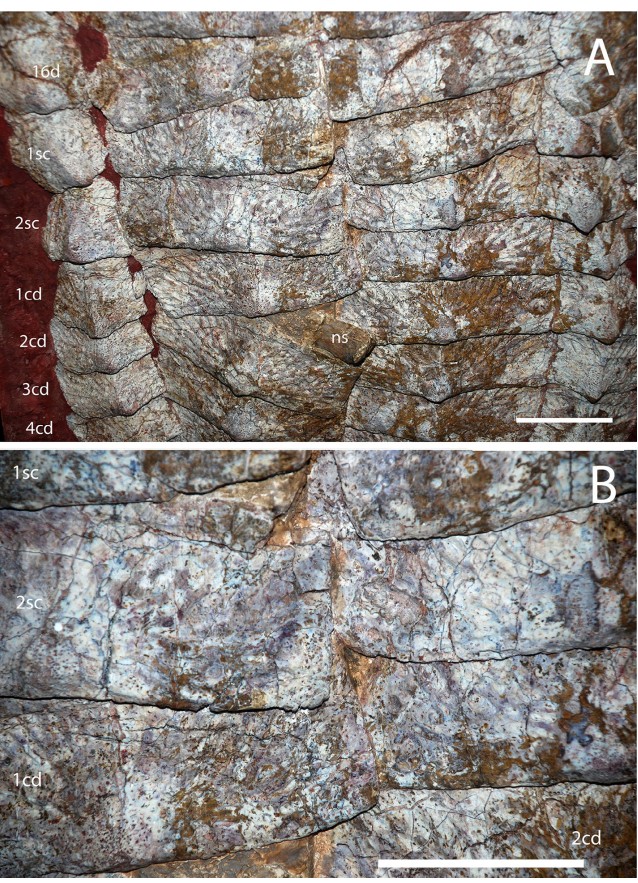

**Figure 15 Holotype osteoderms of *Calyptosuchus wellesi*.** (A, B) Close-ups of the carapace of the holotype of *Calytosuchus wellesi* (UMMP 13950) showing details of the paramedian osteoderms. (A) Sacral and anterior caudal region in dorsal view. (B) Close-up of last sacral and first caudal rows in dorsal view. Note lack of raised posteromedial boss. d, dorsal trunk row; sc, sacral row; cd, caudal row. Scale bars equal 10 cm.

ridges are more anastomosing. *C. wellesi* also lacks the elongate parallel grooves and ridges found in *A. scagliai* (PFV 2073). Furthermore, the posteromedial corners of the paramedians are flat and ornamented, lacking the distinct raised triangular boss of *S. deltatylus* (PEFO 34616) or the triangular unornamented area of *A. eisenhardtae* (PEFO 34638). The lateral edge here is slightly indented for a short triangular process of the lateral osteoderm, but is not deeply "cut-off" as in typothoracines such as *Paratypothorax* sp. (PEFO 3004) or as in *A. eisenhardtae* (PEFO 34638).

Isolated osteoderms from the *Placerias* Quarry (Figs. 16A–16K) demonstrate that at least some of the dorsal trunk paramedians had a weakly developed ventral strut (e.g., UCMP 136744; Figs. 16B, 16D and 16E), an anterolateral projection (e.g., UCMP 126846; Fig. 16F), "scalloping" of the medial portion of the anterior bar (e.g., UCMP 136744, UCMP 126844, UCMP 126801; Figs. 16G, 16H and 16J), and a distinct anteromedial projection (UCMP 136744, UCMP 126844, MNA V2930; UCMP 126801; Figs. 16G–16J). Some of the osteoderms (e.g., UCMP 136744; Figs. 16C–16E) are strongly flexed ventrally. Osteoderms from smaller, presumably less mature individuals, have

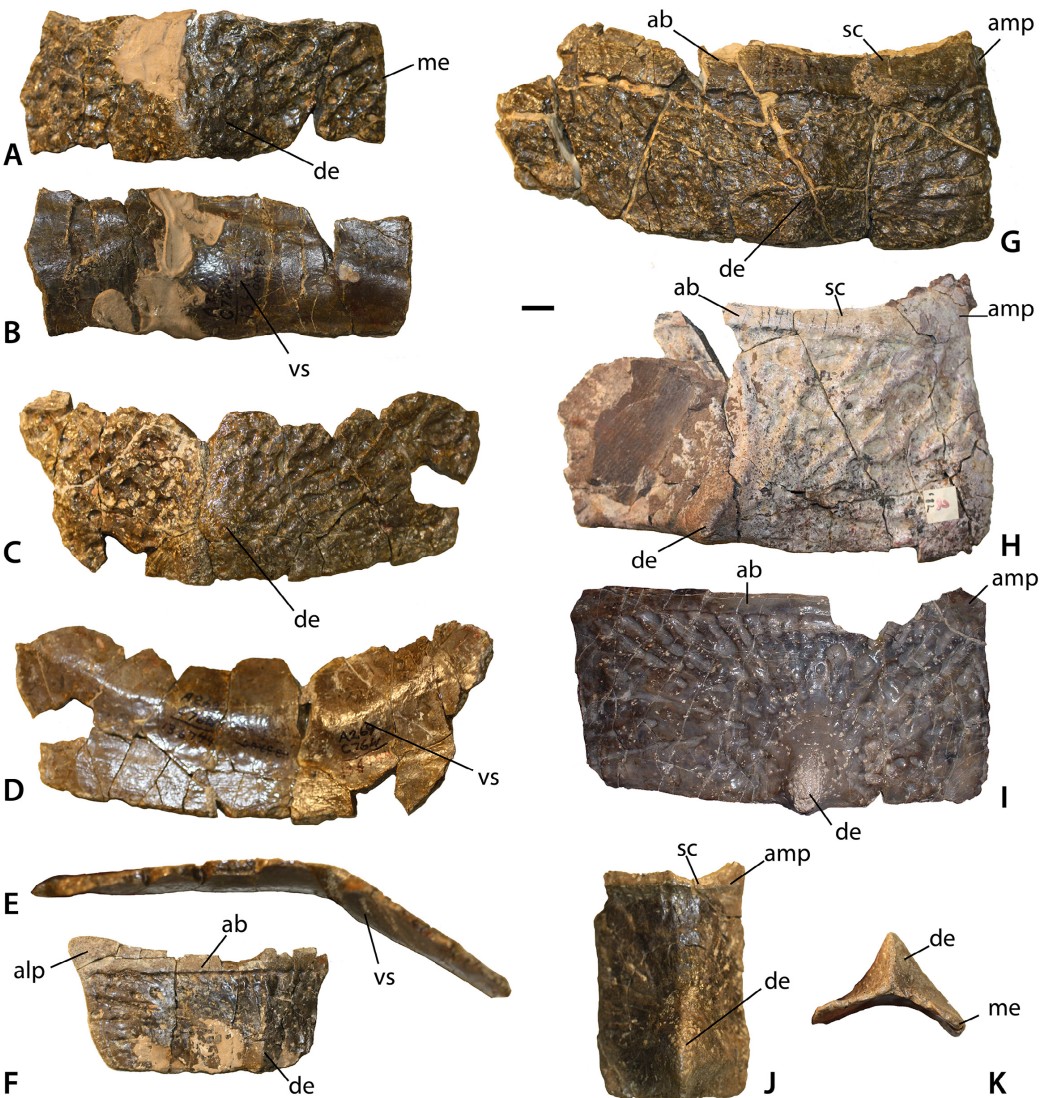

**Figure 16 Paramedian osteoderms of *Calyptosuchus wellesi.*** Paramedian osteoderms of *Calyptosuchus wellesi.* (A, B) UCMP 136744, left anterior dorsal trunk osteoderm in dorsal (A) and ventral (B) views. (C–E) UCMP 136744, right posterior dorsal trunk osteoderm in dorsal (C), ventral (D), and anterior (E) views. (F) UCMP 126846, left dorsal trunk osteoderm in dorsal view. (G) UCMP 136744, left dorsal mid-trunk osteoderm in dorsal view. (H) UCMP 126844, left dorsal mid-trunk osteoderm in dorsal view. (I) MNA V2930, left posterior dorsal trunk osteoderm in dorsal view. (J. K) Left posterior mid-caudal osteoderm in dorsal (J) and posterior (K) views. Scale bar equals 1 cm. ab, anterior bar; alp, anterolateral process; amp, anteromedial process; de, dorsal eminence; me, medial edge; sc, scalloped area of anterior bar; vs, ventral strut.

dorsal eminences in the form of elongate keels rather than blunt pyramidal bosses. This is similar to the condition in smaller sized taxa such as *A. ferratus* (*Schoch, 2007*) and *A. scagliai* (PVL 2073).

Closer to the end of the tail the paramedian osteoderms become longer than wide with strong pyramidal dorsal eminences (e.g., UCMP 126801; Figs. 16J and 16K). Even more distally, the bosses become reduced and blunter, but the osteoderms thicken

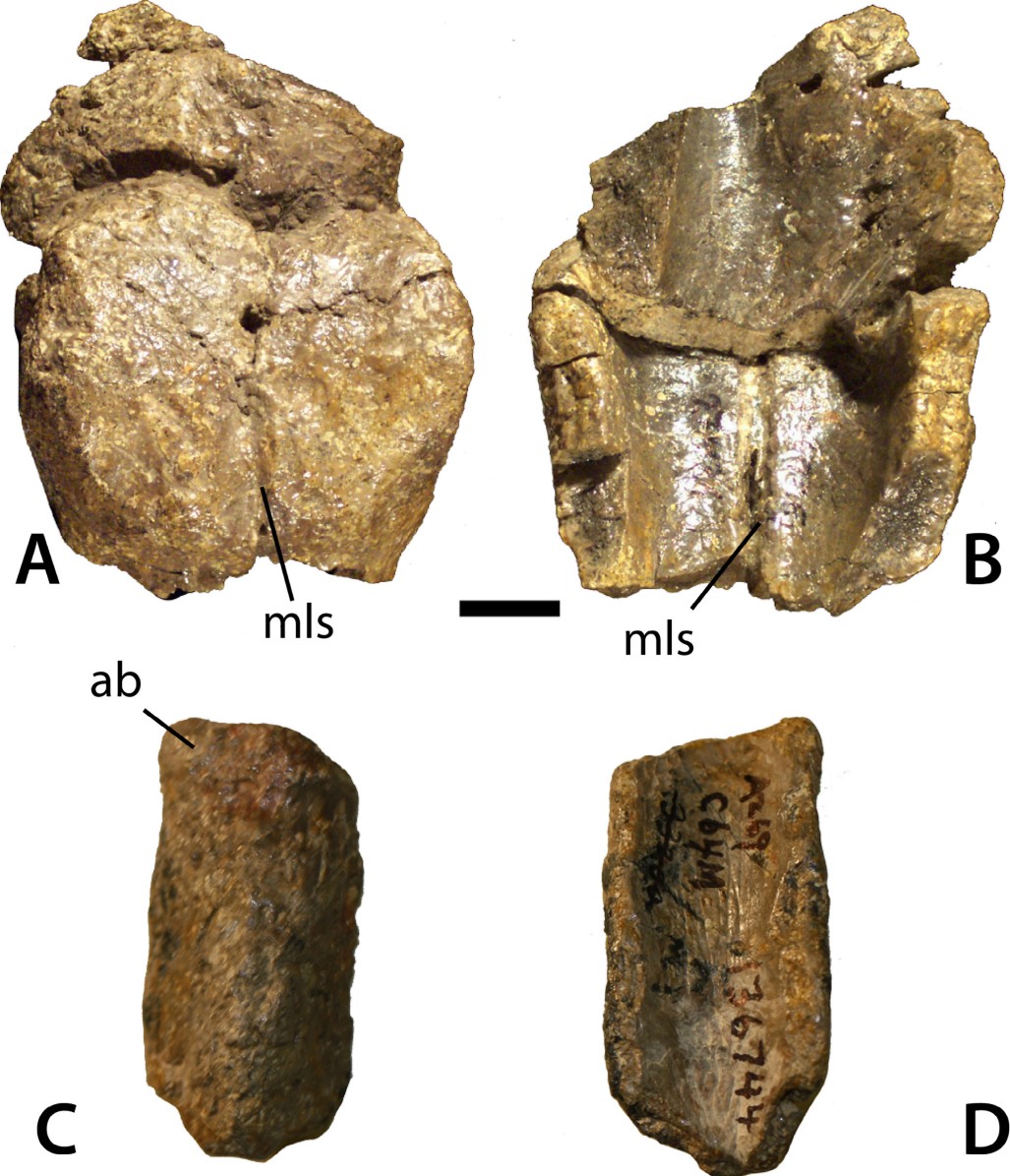

**Figure 17 Caudal paramedian osteoderms of *Calyptosuchus wellesi.*** Distal caudal paramedian osteoderms of *Calyptosuchus wellesi* (UCMP 136744). (A, B) Two semi-articulated sets of fused paired osteoderms in dorsal (A) and ventral (B) views. (C, D) Isolated osteoderm in dorsal (C) and ventral (D) views. Scale bar equals 1 cm. ab, anterior bar; mls, mid-line suture.

significantly and in some cases start to fuse to each other (e.g., UCMP 136744; Figs. 17A–17D). This is very similar to the condition in *S. deltatylus* (PEFO 34045).

### Lateral osteoderms

The lateral osteoderms from the ninth dorsal trunk row (of 16 total) through the 16th caudal rows (of approximately 40 according to *Schoch (2007)* for *A. ferratus*) are present and well-preserved in the holotype (UMMP 13950). Thus, the positions of

isolated lateral osteoderms with matching anatomy can be placed with confidence. Aetosaurian lateral osteoderms are roughly square to rectangular with a pronounced dorsal eminence or boss (*Heckert & Lucas, 2000*). Typically the osteoderms are flexed to some degree, divided into two "flanges" (dorsal and lateral or ventral) by the eminence (*Long & Ballew, 1985*; *Parker, 2007*). Importantly, all of the lateral osteoderms in UMMP 13950 have more rectangular dorsal flanges, however, lateral osteoderms with strongly triangular dorsal flanges are present in the referred material of *C. wellesi*. These osteoderms must be from positions anterior to the ninth dorsal row. All of the lateral osteoderms have prominent anterior bars, pyramidal dorsal eminences, and a surface ornamentation of grooves and ridges radiating from the eminence.

The anteriormost lateral osteoderms of the trunk series are well-represented in specimen UCMP 27225, a partial skeleton represented by osteoderms and vertebrae and collected by Charles Camp near St. Johns in 1926. They are quadrilateral in dorsal view with distinct dorsal and lateral flanges separately by an elongate keeled dorsal eminence with a pyramidal terminal end that projects just slightly beyond the posterior osteoderm margin (Figs. 18A–18D). The dorsal flange is distinctly triangular in dorsal view and is reduced in size compared to the lateral flange. The lateral flange appears to increase in width in more posteriorly situated osteoderms. The medial edge of the dorsal flange is strongly sigmoidal and the anterior bar is indented where the anterolateral projection of the adjacent paramedian osteoderm overlies it.

In the next positions, but still anterior to the ninth dorsal trunk row, the dorsal flanges retain their sigmoidal lateral edge, but become more quadrilateral in dorsal view (Figs. 18E and 18F). The lateral flanges are very wide and rectangular. They are still significantly larger than the dorsal flange. The next form of lateral osteoderm occur in the 9th–12th dorsal trunk positions based on comparison with the holotype (UMMP 13950) and are best represented in the *Placerias* Quarry material by left and right osteoderms (UCMP 136744; Figs. 18G–18J).

The dorsal eminence is larger and very hook-like. The dorsal flange is quadrilateral in dorsal view and maintains the strongly sigmoidal medial margin. The lateral flanges are still much wider than the dorsal flanges but are no longer rectangular. Instead they are strongly quadrilateral with a distinct mediolateral slant so that the anterior margin is much wider than the posterior margin. This forms a distinct anterolateral "wing" that characterizes the osteoderms from this portion of the carapace. In posterior view the angle between the flanges is approaching 90°, much more flexed than the preceding lateral osteoderms.

The sacral and anteriormost caudal lateral osteoderms are represented by a right (UCMP 78751) and two left (UCMP 136744, MNA V3744) osteoderms (Figs. 18K–18N). These osteoderms are reduced in overall width, the lateral flange remains larger than the dorsal flange, but only slightly and anterolateral "wing" is no longer prominent. The dorsal eminence is still strong, but not as hook-like as the previous osteoderms.

At about the third caudal row the dorsal eminence of the lateral osteoderms becomes very rectangular, and the dorsal and lateral flanges are more equal in size. Overall the osteoderms are lengthening anteroposteriorly, corresponding with the increasing length of the caudal vertebrae. These positions are represented by two right osteoderms, UCMP 27048 from

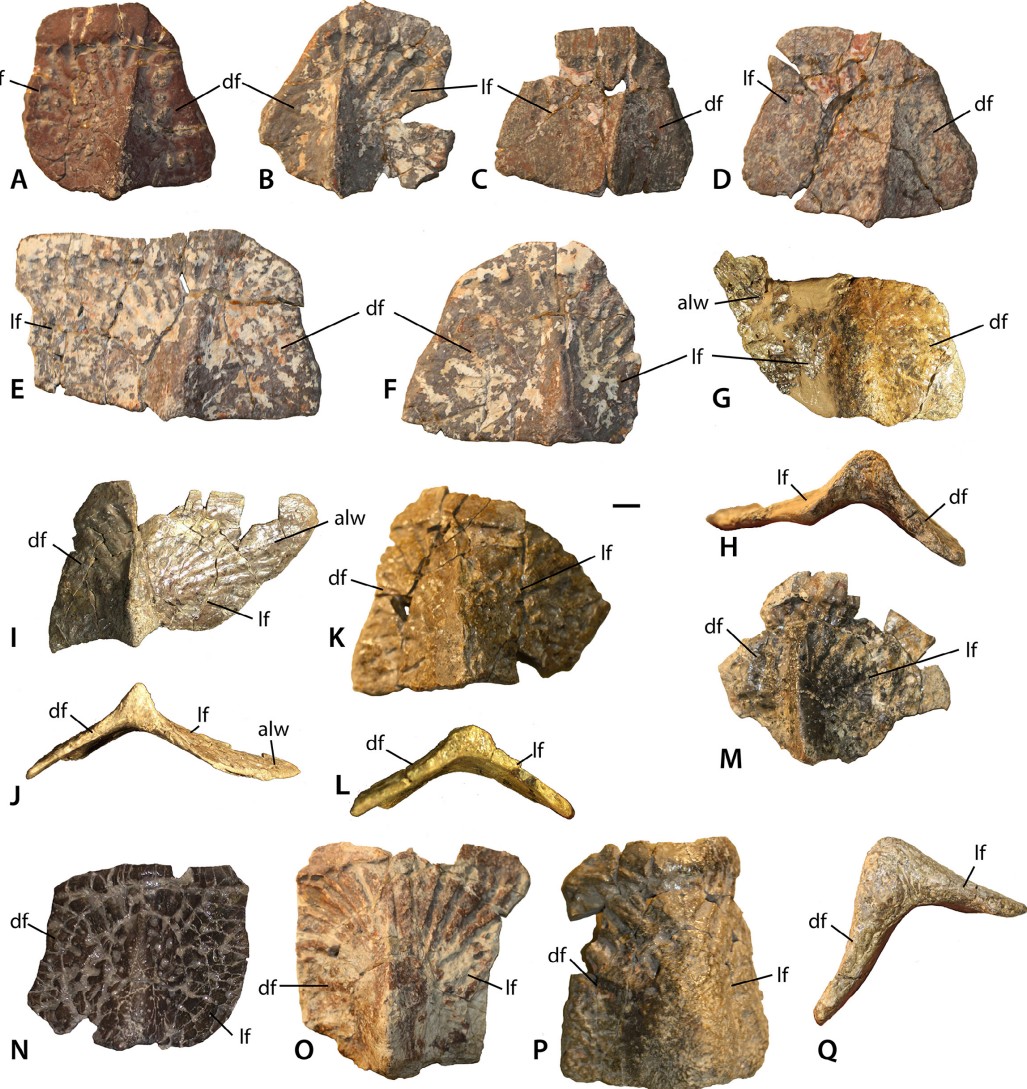

**Figure 18 Lateral osteoderms of *Calyptosuchus wellesi.*** Lateral osteoderms of *Calyptosuchus wellesi.* (A–D) anteriormost dorsal trunk lateral osteoderms (UCMP 27225) from the left (A, C, D) and right (B) sides in dorsal view. (E, F) anterior dorsal trunk lateral osteoderms (UCMP 27225) from the left (E) and right (F) sides in dorsal view. (G–J) Posterior dorsal trunk lateral osteoderms (UCMP 136744) from the left (G, H) and right (I, J) sides in dorsal (G, I) and posterior (H, J) views. (K–N) Sacral and anteriormost caudal lateral osteoderms (UCMP 78751, K–L; UCMP 136744, M; MNA V3744, N) of the right side in dorsal (K, M, N) and posterior (L) views. (O–Q) Anterior-mid-caudal lateral osteoderms (UCMP 27048, O; UCMP 136744, P, Q) of the right side in dorsal (O, P) and posterior (Q) views. Scale bar equals 1 cm. alw, anterolateral 'wing'; df, dorsal flange, lf, lateral flange.

the Blue Hills area of St. Johns, and UCMP 136744 from the *Placerias* Quarry (Figs. 18O–18Q). The dorsal eminence is taller but blunter, not hook-like. The angle of flexion between the dorsal and lateral flanges is a strong 90° in these osteoderms.

### *Ventral osteoderms*

Ventral trunk osteoderms are best represented in UCMP 27225 (Fig. 19). They are square to broadly rectangular with a strong, but narrow anterior bar. The external surface

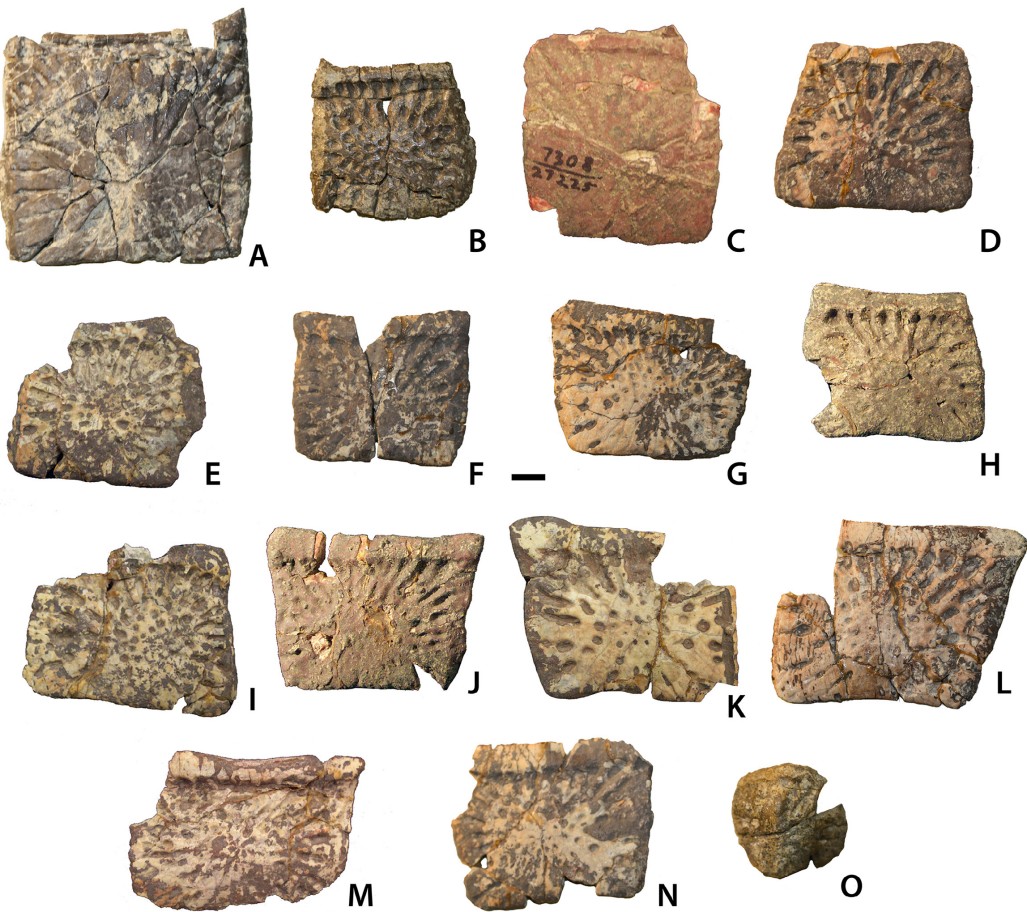

**Figure 19 Ventral and appendicular osteoderms of *Calyptosuchus wellesi*.** Ventral and appendicular osteoderms of *Calyptosuchus wellesi*. (A) UCMP 175148, ventral osteoderm in ventral view. (B) UCMP 136744, ventral osteoderm in ventral view. (C–N) UCMP 27225, ventral osteoderms in ventral view. (O) UCMP 136744, external surface of an appendicular osteoderm. Scale bar equals 1 cm.

ornamentation consists of a fine pattern of grooves and ridges radiating from a central, unraised area on the osteoderm.

### *Appendicular osteoderms*

Numerous appendicular osteoderms are preserved close to life position in the holotype (UMMP 13950: Fig. 13). They consist of small rounded to oval osteoderms with faint surface pitting. They would have been situated manly along the upper portion of the individual limbs.

## DISCUSSION

### Phylogenetic relationships of *C. wellesi*

*Calyptosuchus wellesi* has been considered one of the better known aetosaurian taxa from the American Southwest. However, it has never been completely described and, whereas our knowledge of many of the other southwestern taxa (e.g., *D. spurensis*, *T. coccinarum*) has increased because of the recovery of new specimens, hardly any new material of

*Calyptosuchus* has been discovered in recent years. Several partial skeletons mentioned by *Parker & Irmis (2005)* and *Parker & Martz (2011)*, including cranial material, are instead referable to a new taxon *Scutarx deltatylus Parker 2016a*. Thus, the best sources of character information on *C. wellesi* are the numerous osteoderms and endoskeletal elements from the *Placerias* Quarry. Unfortunately past assignments (*Long & Murry, 1995*) of this material to various taxa are problematic because no methodology for assigning material from the quarry to various taxa was discussed. I have attempted here to use the only source of data remaining from the original excavations, the grid numbers, to look for clues regarding possible association of endoskeletal elements with the diagnostic osteoderms, however, in many cases the data are unequivocal because of the mixture of osteoderms of more than one aetosaurian taxon and because the original workers did not collect the majority of the osteoderms from the east side of the quarry.

The approach taken in this study follows previous workers (*Camp & Welles, 1956*; *Long & Murry, 1995*) in that there are only two significant aetosaurian taxa known from the *Placerias* Quarry, *C. wellesi* and *D. spurensis*. Where possible, elements have been assigned based on the direct association of these elements (e.g., dentary, cervical vertebrae) with diagnostic osteoderms, as well as through association of elements (e.g., pelvis and femur, osteoderms, trunk vertebrae) that can be unambiguously assigned to *Calyptosuchus* using apomorphies that distinguish them from *Desmatosuchus*. Other assignments of bones to *Calyptosuchus* are more problematic such as the tibia and fibula, which *Murry & Long (1989)* and *Long & Murry (1995)* differentiated from *Desmatosuchus* by assigning the more "gracile" elements to the smaller, and thus in their opinion, presumably more "gracile" *Calyptosuchus*. The possibility that these differences represent sexual dimorphism in a single taxon was not considered by those authors. Difference in size of elements has been proposed as sexual dimorphic traits for aetosaurs such as *S. robertsoni* (*Walker, 1961*) and *L. meadei* (*Elder, 1978*), but this is very difficult to evaluate with the present sample sizes of North American aetosaurs (*Parker & Martz, 2010*) and without an independent confirmation of sex, body size is rarely a reliable indicator of sexual dimorphism in extinct vertebrates. Aetosaurian postcrania are fairly rare in comparison to osteoderms (*Desojo et al., 2013*) and determinations of variation because of sexual dimorphism cannot be made, but should be considered a possibility although more complete finds are required to clarify. In recent years the east side of the quarry, as well as the nearby Downs Quarry (MNA 207-2; *Jacobs & Murry, 1980*), has been reopened by crews from the North Carolina State Museum and Appalachian State University. Results are still forthcoming, but hopefully these sites will prove rich in associated remains of *Calyptosuchus* and help further clarify the osteology of this taxon.

Presently *C. wellesi* lacks discrete autapomorphies, but can be diagnosed using a unique combination of characters including the presence of a ventral strut on the dorsal paramedian trunk osteoderms and large posteriorly situated dorsal eminences as in typothoracisins; paramedian osteoderms with a strongly raised anterior bar with a "scalloped" anterior edge and distinct anteromedial and anterolateral projections as in non-desmatosuchin desmatosuchians and in aetosaurines; a radial pattern of grooves and ridges on the dorsal paramedian osteoderms as in non-desmatosuchin aetosaurs;

the lack of a raised triangular boss in the posteromedial corner of the paramedian osteoderms as in *S. deltatylus*; the lack of a smooth triangular patch of bone in the posteromedial corner of the paramedian osteoderms as in *A. eisenhardtae* and *S. robertsoni*; and a squared off posterior end of the iliac blade as in *A. scagliai*.

Scoring of the majority of these character states into a phylogenetic analysis was completed by *Parker (2016a)*. The resulting strict consensus tree of 201 steps from that study (Fig. 20) recovered *C. wellesi* as a non-desmatosuchin desmatosuchine and the sister taxon of *A. eisenhardtae* + *S. deltatylus*. Although the paramedian and lateral osteoderms of these three taxa are very similar, *C. wellesi* differs from the other two in lacking an unornamented mediolateral corner on the trunk paramedian osteoderms (*Parker, 2016a*, *2016b*). Once assigned to the genus *Stagonolepis* (*Murry & Long, 1989*), *Calyptosuchus* is not recovered in a *Stagonolepis* clade with *S. robertsoni* or *S. olenkae* (Fig. 20; *Parker, 2016a*), thus it is maintained here as a distinct monotypic genus.

## Problems with genus-level taxa in vertebrate paleontology

*Murry & Long (1989)* assigned *C. wellesi* to the genus *Stagonolepis* without explanation, but presumably based on similarities of the osteoderms, and this has been followed in many aetosaur studies (*Long & Murry, 1995*; *Heckert & Lucas, 1999*; *Heckert & Lucas, 2000*; *Parker, 2007*); however, comparisons with the material of *S. robertsoni* and optimization of characters states in a phylogenetic context (*Parker, 2016a*) suggest that many of the similarities of the osteoderms in *C. wellesi* and *S. robertsoni* are plesiomorphic for Aetosauria including the dorsal radial patterning, raised anterior bar, and medially-offset dorsal eminence, which are also found in the non-stagonolepidid aetosaurian *A. scagliai* (*Desojo, Ezcurra & Kischlat, 2012*; *Heckert et al., 2015*; *Schoch & Desojo, 2016*; *Parker, 2016a*). Other differences found in *Calyptosuchus* and not *Stagonolepis* include a squared off posterior end of the iliac blade, transversely oval articular faces of the cervical vertebrae, and elongate anterolateral projections of the anterior bar on the paramedian osteoderms (*Parker, 2016a*).

To some workers this may not seem enough to separate these two species into different genera; however, other aetosaurs such as *N. engaeus* from South America possess the same plesiomorphic characters of the paramedian armor and much of the postcrania (*Parker, 2014*), yet to date no published study has ever proposed placing that species into the genus *Stagonolepis* (*Heckert & Lucas, 2000*; *Parker, 2007*; *Desojo et al., 2013*). Compounding this issue is the reality that for most of the history of taxonomy, a genus has been nothing more than a Linnaean taxonomic rank used to subjectively compile "morphologically similar" species into a single taxon (*Stuessy, 2009*).

The generic name is possibly the most subjectively-determined rank of the Linnaean taxonomic system (*Clarke, 2004*; *Stuessy, 2009*; *Vences et al., 2010*); however, the current enacted taxonomic codes (e.g., the International Code of Zoological Nomenclature [ICZN]) require establishment of a Linnaean binomen includes a distinct genus name. Yet, despite the voluminous amount of published literature dedicated to the "species problem" (see *Mayden, 1997*; *Wiens, 2004*; *de Queiroz, 2007* and references therein), comparatively little has been written regarding concepts on how to delimit genera.

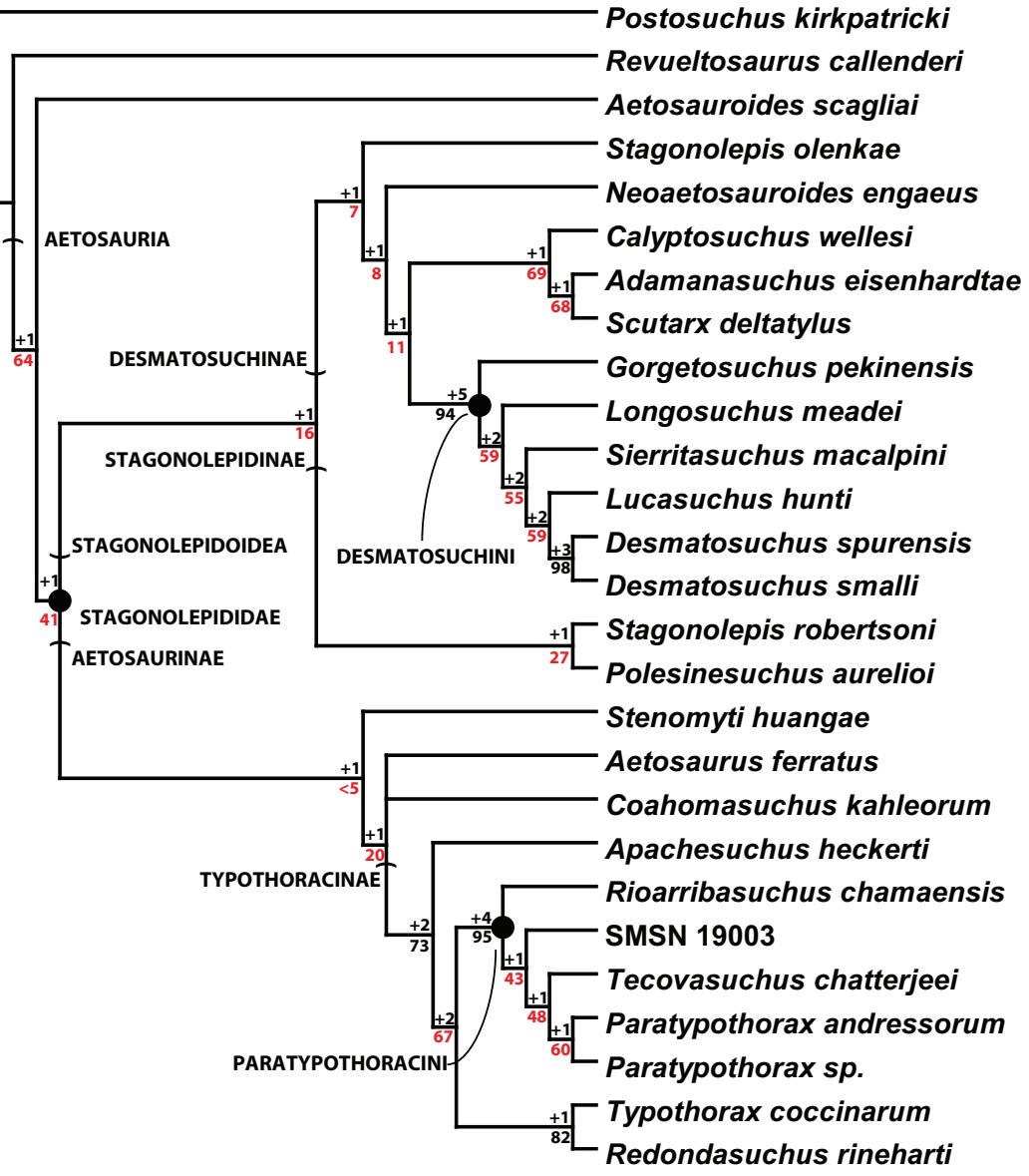

**Figure 20 Cladogram of aetosaurian relationships.** Strict consensus cladogram of 3 MPTs depicting hypothesized phylogenetic relationships of the Aetosauria. Bremer support (black) and bootstrap support (red) values are provided for nodes. Reproduced from *Parker (2016a)*.

*Gill, Slikas & Sheldon (2005)* argued that to be descriptively useful a genus-group taxon should be (1) monophyletic, (2) reasonably compact (i.e., not containing too many species-group taxa), and (3) ecologically, morphologically, or biologically distinct. These last two points fit well with the traditional view of a genus as an assemblage of species that have more significant features in common amongst themselves then with any other species (i.e., they can be diagnosed; *Rowe, 1988*; *Stuessy, 2009*). A review of a set of volumes of the Journal of Vertebrate Paleontology from 2010 demonstrates that many vertebrate paleontologists accept the first point, that genera should be monophyletic

(i.e., they can be defined; *Rowe, 1988*), and that the discovery of paraphyletic genera in a phylogenetic analysis may require the formulation of new taxonomic names at the genus-level (*Lyson & Joyce, 2010*; *Maxwell, 2010*; *Cadena, Bloch & Jaramillo, 2010*). However, this approach tends to result in the establishment of monotypic genera (*Lyson & Joyce, 2010*; *Cadena, Bloch & Jaramillo, 2010*), which have been considered problematic by some workers (*Platnick, 1976*, *1977a*, *1977b*; *de Queiroz & Gauthier, 1992*; *Loeuille, Sinischalchi & Pirani, 2014*), especially in estimating diversity of different vertebrate groups through time.

Monotypic genera have also been criticized as redundant because they offer no information regarding phylogenetic relationships at the genus-level in that they do not provide an operational name for a clade of terminal taxa (*de Queiroz & Gauthier, 1992*; *Lee, 2003*; *Dayrat et al., 2008*). In a phylogenetic study utilizing only terminal taxa at the genus-group level, the structure of branching events in the phylogenetic analysis requires that in the outermost nodes of the recovered tree each terminal taxon should have a sister taxon at roughly the equivalent taxonomic level having originated in the same cladogenetic event. Thus, it appears fair to assume that if genera are to be treated as clades, then all of the species within these clades should be provided the same genus-level name (*Clarke, 2004*; *Lyson & Joyce, 2010*; *Stocker, 2013*). However, choosing the node at which to define these genera is subjective (hypothetically all of Aetosauria could represent a single genus) and extreme care must be taken that this is not done based on overall similarity. This is of extreme importance because the genus-group level is often the taxonomic level utilized in higher level Triassic vertebrate paleontology studies exploring biostratigraphy, biochronology, biogeography, phylogeny, and extinction (*Benton, 1983*; *Lucas, 1998*; *Brusatte et al., 2008*; *Ezcurra, 2010*; *Stocker, 2010*; *Nesbitt, 2011*; *Parker & Martz, 2011*). Thus it is important that genus-level taxa are not only monophyletic, but also that they only define stable clades based on discrete apomorphies (*Padian, Lindberg & Polly, 1994*; *Angielczyk & Kurkin, 2003*; *Vences et al., 2010*).

Monotypic genera also tend to indicate unclear relationships between species through a lack of synapomorphies (i.e., the monotypic taxon is highly autapomorphic) or a lack of resolution between a group of taxa (i.e., polytomous phylogenetic relationships) (*Schrire & Lewis, 1996*; *Loeuille, Sinischalchi & Pirani, 2014*). However, when first developed, the purpose of the genus-level rank was to serve as a means to group what were hypothesized to be closely-related species. With the advent of phylogenetic systematics this role is no longer required as it is the recovered phylogenetic trees that hypothesize and define relationships, not the *a priori* assigned genus rank based on character diagnoses. Genera are discerned by character differences; however, relationships are defined by shared characteristics, so autapomorphic specimens that do not fit readily into existing monophyletic groups (i.e., genus-level terminals) should be coded separately in phylogenetic analyses, so that their relationships can be tested *a posteriori* (*Schrire & Lewis, 1996*). In cases where recovered genus-level clades are unstable and the exact internal relationships ambiguous, it is probably best to erect monospecific taxa to promote taxonomic stability of the binomen and eliminate the ambiguity caused by frequent shifting of species within genera (*Martz & Small, 2006*; *Vences et al., 2010*).

This in turn can provide clarity to and avoid compounded analytical mistakes in higher-level studies that utilize supraspecific taxa (e.g., biostratigraphy and biogeography).

Within Aetosauria, *S. deltatylus* appears to share the most anatomical features with *C. wellesi* (*Parker, 2016a*, *2016b*). The phylogenetic analysis from that study (reproduced here as Fig. 20) supports a close relationship between *S. deltatylus* and *A. eisenhardtae*; however, it also demonstrates that as the sister taxon to *Scutarx* + *Adamanasuchus*, *C. wellesi* is also very closely related. Therefore, it is plausible that these three species could all be assigned to the genus *Calyptosuchus*, as this is the oldest valid genus-level name available of the three. However, overall clade support is weak and consideration of the results recovered from past studies that provide modifications to existing phylogenies of the Aetosauria (*Desojo, Ezcurra & Kischlat, 2012*; *Heckert et al., 2015*) strongly demonstrates that future modifications to character scoring or the addition of new taxa could significantly alter the constituency of this clade and the position of those individual taxa. Shifting species between genera based on developing phylogenetic hypotheses is not encouraged because it promotes taxonomic instability at the genus-level (*Pauly, Hillis & Cannatella, 2009*; *Langer, da Rosa & Montefeltro, 2017*).

The delimitation of species and genera in vertebrate paleontological studies is clearly an epistemological problem, because it is extremely unlikely that two recognized terminal sister taxa actually represent their respective evolutionarily closest relatives in life. The incompleteness of the fossil record provides the strong possibility that another taxon could eventually be found that could split existing recovered sister-taxa even in the purportedly best-supported phylogenetic hypotheses (e.g., Aphanosauria, *Nesbitt et al., 2017*). Even individual specimens, because of incompleteness, cannot be unambiguously assigned to an existing species in many cases because each individual specimen could represent a previously unrecognized sister taxon instead. Thus, monotypic genera can provide a conservative approach to taxonomic stability.

## CONCLUSION

Use of quarry data from the collection of *Calyptosuchus* material from the *Placerias* Quarry of Arizona allows for hypotheses to be made regarding the assignment of non-osteoderm material to this taxon. Furthermore, a previously undescribed specimen (UCMP 27225) allows for the referral of the first unambiguous skull material (dentary) to be assigned to this taxon. Although it presently has no discrete autapomorphies, *C. wellesi* can be diagnosed by a unique combination of characters and supported by phylogenetic analysis. Many previous referrals of material to *Calyptosuchus* has been demonstrated to belong to other taxa instead including *A. eisenhardtae*, *S. deltatylus*, and an undescribed Adamanian paratypothoracisin. Despite this *Calyptosuchus* is one of the most common aetosaurians in the Western United States and an index taxon of the early Adamanian Tielzone. The name *Calyptosuchus* is retained and encouraged as the applicable genus name for the species *wellesi* because assignments of taxa to multi-species genus-level names are problematic and in this case provides a proposed taxonomic relationship that cannot be unambiguously supported, even by phylogenetic analyses. Because of the inherent limitations of the fossil record, referral of specimens and species

to species and genera respectively is an epistemological problem in Triassic vertebrate paleontology. The preferred use of monotypic genera such as *C. wellesi* can promote taxonomic stability in ever-changing hypotheses of clades.

## INSTITUTIONAL ABBREVIATIONS

**AMNH FR**  Frick Collection, American Museum of Natural History, New York, NY, USA
**MNA**  Museum of Northern Arizona, Flagstaff, AZ, USA
**PEFO**  Petrified Forest National Park, Arizona, USA
**PVL**  Paleontología de Vertebrados, Instituto "Miguel Lillo," San Miguel de Tucumán, Argentina
**TMM**  Texas Vertebrate Paleontology Collections, University of Texas, Austin, Texas, USA
**TTU**  The Museum at Texas Tech, Lubbock, TX, USA
**UCMP**  University of California Museum of Paleontology, Berkeley, CA, USA
**UKNHM**  The Natural History Museum, London, UK
**UMMP**  University of Michigan Museum of Paleontology, Ann Arbor, MI, USA

## ACKNOWLEDGEMENTS

This manuscript was originally part of a PhD dissertation from the University of Texas at Austin. Timothy Rowe, Christopher Bell, Julia Clarke, Sterling Nesbitt, and Hans-Dieter Sues provided comments on this earlier version of the manuscript. Discussions with Michelle Stocker, Adam Marsh, and Sterling Nesbitt helped formulate some of the ideas presented here about species and genera. Thank you to Pat Holroyd, Mark Goodwin, and Kevin Padian (UCMP) for access to material, researcher notes, and for discussions. Thanks also to the late Gregg Gunnell (then at UMMP), David Gillette (MNA), Janet Gillette (MNA), Angela Milner (UKNHM), and Lorna Steele (UKNHM) for access to specimens under their care. David Gower (UKNHM) shared Alick Walker's photographic collection, notes, and specimens. Photos of material were supplied by Randall Irmis and Jeffrey Wilson. Thank you to J. Michael Parrish, an anonymous reviewer, Adam Marsh, Sterling Nesbitt, and academic editor Graciela Piñeiro for their thoughtful reviews that improved the manuscript. This is Petrified Forest National Park Paleontological Contribution No. 53. Any opinions, findings, or conclusions of this study represent the views of the author and are not those of the U.S. Federal Government.

### Funding

This project was supported by the Francis L. Whitney Endowed Presidential and the Ernest and Judith Lundelius scholarships from the University of Texas at Austin. The Systematics Association supported travel. There was no additional external funding received for this work. The funders had no role in study design, data collection and analysis, decision to publish, or preparation of the manuscript.

## Grant Disclosures

The following grant information was disclosed by the authors:
Francis L. Whitney Endowed Presidential and the Ernest and Judith Lundelius scholarships from the University of Texas at Austin.

## Competing Interests

The author declares that he has no competing interests.

## Author Contributions

- William G. Parker conceived and designed the experiments, performed the experiments, analyzed the data, contributed reagents/materials/analysis tools, wrote the paper, prepared figures and/or tables.

## Data Availability

The paleontological specimens described in this manuscript are held at the University of California Museum of Paleontology at Berkeley, California and at the University of Michigan Museum of Paleontology at Ann Arbor, Michigan.

## Supplemental Information

Supplemental information for this article can be found online at http://dx.doi.org/10.7717/peerj.4291#supplemental-information.

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
