# Peer review of "Redescription of Calyptosuchus (Stagonolepis) wellesi (Archosauria: Pseudosuchia: Aetosauria) from the Late Triassic of the Southwestern United States with a discussion of genera in vertebrate paleontology"

_PeerJ, doi:10.7717/peerj.4291_

## Round 0.1 · original submission · Minor Revisions

Dear Dr. Parker,

We have already two review reports about your manuscript and both conclude that it should be published by PeerJ. They recommended some changes however, that is important that you address.

I was also wondering that given that two taxa can be identified from osteoderm morphology, but you can find just isolated skeletal elements (e.g. vertebrae, humerus, etc.) if you considered the possibility of sexual dimorphism to explain some minor anatomical differences that you find among the isolated bones or you do consider that they can certainly demonstrate the presence of the two taxa. In the last case, I would like to see a more detailed argumentation.

Hopping to see the revised version of your manuscript submitted soon.

Best regards,
Graciela Piñeiro

·

Basic reporting

No Comment

Experimental design

This manuscript offers a useful compilation and description of material referred to
Calyptosuchus wellesi, based primarily on the holotype and referred material from the UCMP collections. The chief deficiency of the analysis is the lack of a clear phylogenetic context. Aetosaur material is notoriously difficult to analyze phylogenetically because so much of the material is fragmentary, and the durable scutes are often the only elements present. Also, there is no clear picture of how individual variation and ontogeny might affect some of the characters used for aetosaur taxonomy, particularly of the vertebrae and scutes. Still, it is important to advance a clear argument as to the systematic arguments put forth. Since the manuscript reverses the recent classification of wellesi as belonging to the genus Stagonolepis, a differential diagnosis should be provided demonstrating the preferred generic assignment of the species based on a differential diagnosis.

Validity of the findings

As described, the specific assignment of C. wellesi is not based on any autapomorphies, but could be supported by a unique combination of synapomorphies present in other taxa. Again, this should be done in the context of a phylogenetic analysis.

Additional comments

One important aspect of this paper is the analysis of the locations of the disarticulated elements that make up Camp’s collection of aetosaur material from the Placerias quarry. These data do offer compelling evidence for the differentiation of material of Desmatosuchus spurensis and C. wellesi from the quarry.

In summary, this is a useful description of important aetosaur material from the Triassic of the southwest. However, in its current form, the paper does not provide a compelling systematic argument for the stated systematic assignments, as opposed to differing arguments made in recent papers. I would suggest that a revised analysis of this type would be a necessary prerequisite for publication. There are abundant recent papers covering phylogeny of aetosaurs, most notably Desojo et al 2013 (for which Parker is a co-author), from which the phylogenetic context for the systematic arguments could be derived.

Editorial Comments regarding citations:

Lines 62, 146 Parker 2005a should be Parker 2005

Lines 95, 98, 100, 102. Should be consistent in citing Fiorillo et al (2000) vs listing all three authors.

Line 106. Irmis et al 2011 not in references.

Lines 183, 184 should be Gregory 1953 not 1953a.

I could not find citations for Ludekker 1887, Heckert and Lucas, 2002, and Heckert et al 2005

Reviewer 2 ·

Basic reporting

It is not sufficient to give a map of the bonebed only (Fig. 1).

Fig. 1 may be improved by adding a geological map of the study area showing the fossil locality and a litholog showing stratigraphic position of the fossil-bearing horizon.

Experimental design

Although it is difficult to separate out materials of Calyptosuchus from other aetosaurian remains, the author has done a commendable job of bringing out the morphological distinctiveness of Calyptosuchus remains.

However, a comprehensive phylogenetic analysis to ascertain the position of Calyptosuchus with respect to other aetosaurs would have much improved the manuscript and strengthened the assertion that ‘Calyptosuchus may be distinguished from other aetosaurs based on unique combination of characters’.

Validity of the findings

No comment.

Additional comments

The manuscript redescribes a medium sized desmatosuchian aetosaur Calyptosuchus wellesi collected from a multitaxic fossil locality, which have yielded other aetosaurian remains. The manuscript is well written, thorough, imparts new information regarding the postcrania of Calyptosuchus, and should be published.

Manuscript may be shortened considerably. It may be reduced by removing sections that are imparting no new information. For example, the paragraphs on astragalus and calcaneum may be reduced to one/two sentences as there are no new morphological information.

---

## Round 0.2 · Minor Revisions

Dear author,

In first place I wish to apology for the delay in my decision, I had to attend other activities that occupied all my time.

Concerning your manuscript, I saw with pleasure that you followed the reviewer’s recommendations and improved substantially your work. However, I found some issues that you have to fix before the article is ready for publication. Please, see the attached annotated pdf and consider the suggestions I made on it. Otherwise, submit a rebuttal letter explaining your point.

Hopping that you will find interesting the suggested changes and corrections, I send you my best wishes.

Merry Christmas!
Graciela Piñeiro

---

## Round 0.3 · Minor Revisions

Dear William,

I was suited will all but one of your comments on the rebuttal letter. Please see below:

You said:

“Although the articular faces of the vertebrae appear flat, they are not so the vertebrae are not truly amphiplatyan. Instead they are very slightly concave and therefore I have considered them to be amphicoelous. I’m not familiar with a term for a vertebra that has one nearly amphiplatyan face and one clearly amphicoelous face although that would be the best description.”

So, the dorsal vertebrae are not truly amphiplatyan, but they are neither truly amphicoelus! Thus, I recommend that you explain that in your manuscript to leave the condition clear to the readers; it could be a paragraph similar to as in the rebuttal letter.

Thanks.
Best regards,
Graciela Piñeiro

---

## Round 0.4 · accepted · Accept

Dear Dr. Parker,

I am pleased to announce that your manuscript ‘Redescription of Calyptosuchus (Stagonolepis) wellesi (Archosauria: Pseudosuchia: Aetosauria) from the Late Triassic of the Southwestern United States with a discussion of genera in vertebrate paleontology’ is ready for its publication in PeerJ. This paper will be an important contribution to the knowledge of Triassic archosaurs from the United States, mainly useful for future studies of isolated or disarticulated cranial and postcranial materials, which represent a common taphonomic feature found in continental Permian and Triassic fossiliferous strata.

Congratulations!
Best wishes,
Graciela Piñeiro